# Deduction of Reservoir Operating Rules for Application in Global Hydrological Models

Hubertus M. Coerver[1, 2], Martine M. Rutten[1], and Nick C. van de Giesen[1]

[1]Water Resources, Faculty of Civil Engineering and Geosciences, Delft University of Technology, Delft, Netherlands
[2]UNESCO-IHE Institute for Water Education, Delft, Netherlands

*Correspondence to:* Bert Coerver (b.coerver@unesco-ihe.org)

**Abstract.** A big challenge in constructing Global Hydrological Models is the inclusion of anthropogenic impacts on the water cycle, such as caused by dams. Dam operators make decisions based on experience and often uncertain information. In this study information generally available to dam operators, like inflow into the reservoir and storage levels, was used to derive fuzzy rules describing the way a reservoir is operated. Using an Artificial Neural Network capable of mimicking fuzzy logic, called the Adaptive-Network-Based Inference System, fuzzy rules linking inflow and storage with reservoir release were determined for 11 reservoirs in Central-Asia, the U.S. and Vietnam. By varying the input variables of the neural network, different configurations of fuzzy rules were created and tested. It was found that the release from relatively large reservoirs was significantly dependent on information concerning recent storage levels, while release from smaller reservoirs was more dependent on reservoir inflows. Subsequently, the derived rules were used to simulate reservoir release with an average Nash-Sutcliffe coefficient of 0.81.

## 1 Introduction

Over the last decades, major advances have been made regarding global data availability. Low-resolution hydrologic states from remote sensing and high resolution parameter fields have become available. Combined with the improvements in computational capabilities and data storage, these advances have provided hydrologists the opportunity to pursue the development of high resolution global hydrological models (GHM) like, among others, PCRGLOB-WB (Van Beek and Bierkens, 2009), waterGAP3 (Döll et al., 2009), WBMplus (Wisser et al., 2010a), SWBM (Orth and Seneviratne, 2013), WR3A (van Dijk et al., 2014) and HBV-SIMREG (Beck et al., 2016).

As indicated by Wood et al. (2011), a major challenge in constructing a GHM is the incorporation of human impacts on the terrestrial water cycle, such as operation of reservoirs. Today, almost 40,000 large reservoirs, containing approximately 6,000 km$^3$ of water and inundating an area of almost 400,000 km$^3$, can be found (Takeuchi et al., 2002). Since these reservoirs contain more than three times as much water as stored in river channels and almost one-sixth of the global annual river discharge, they have a significant impact on the timing, volume and peaks of river discharges (Baumgartner and Reichel, 1975). These impacts can have severe environmental consequences. For example, both the drying up of the Aral Sea and the depletion of Lake Urmia

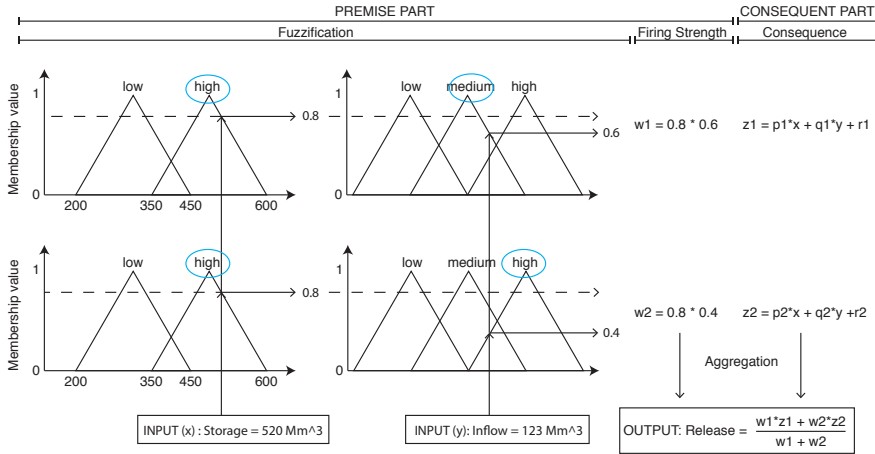

**Figure 1.** An example showing the four steps of fuzzy reasoning.

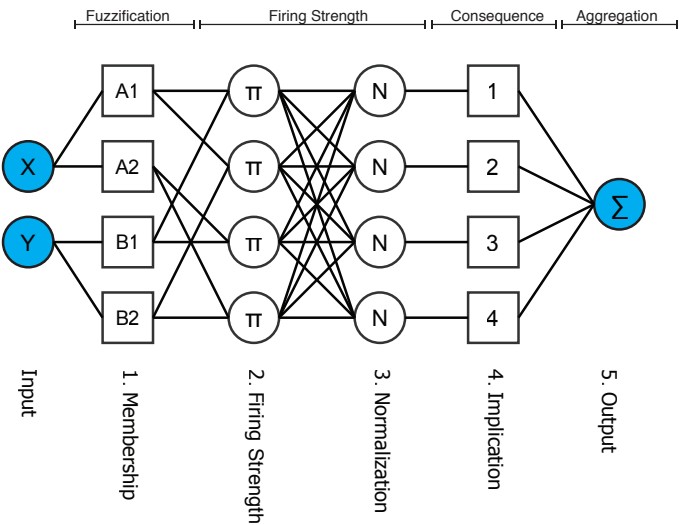

**Figure 2.** The five layers of ANFIS for a network with two input variables and two membership functions per variable. Note that square nodes contain trainable parameters while circular nodes are fixed.

in northern Iran are believed to be results of anthropogenic changes in river flow (Precoda, 1991; Aghakouchak et al., 2015). Implying that in order for GHMs to function properly, the effects of reservoirs have to be incorporated.

Nazemi and Wheater (2015) review the algorithms currently used in GHMs to deal with reservoirs and conclude that large uncertainties remain and there is room for improvement, possibly by representing reservoir operations through rule-based models.

Actual reservoir operation is an imprecise and vague undertaking, since operators always face uncertainties about inflows, evaporation, seepage losses and various water demands which need to be met. They often base their decisions on experience and available information, like reservoir storage and the previous periods inflow (Russell and Campbell, 1996; Hejazi et al., 2008). This study proposes a method to link this information to their decisions.

Fuzzy logic, as introduced by Zadeh (1965), is a popular method to model decision-making processes, that has found its way into reservoir management optimisation models nearly two decades ago (Macian-Sorribes and Pulido-Velazquez, 2016; Russell and Campbell, 1996; Panigrahi and Mujumdar, 2000; Shrestha et al., 1996; Chang and Chang, 2006, 2001; Mousavi et al., 2007). Fuzzy logic has not been used within the field of reservoir release and storage modelling.

In this study, historical inflows, storage-levels and releases are used to derive fuzzy rules that describe the release decisions of dam operators using Artificial Neural Networks (ANN). These rules can be used as the basis for a macro-scale reservoir algorithm. Validity of the derived rules is tested by using them to simulate the reservoirs release and comparing these releases with the actual releases. In order to evaluate if the rules are capable of improving upon the way reservoirs are currently modelled in GHMs, a quantitative comparison is made with a simulation based reservoir algorithm. Additionally, the accuracies of simulated releases resulting from different configurations of the fuzzy rules are compared mutually in order to link the results to the impoundment ratios of the dams.

## 2 Brief review of macro-scale reservoir algorithms

Many macro-scale algorithms, which cannot rely on detailed information on reservoir operation policies used in small-scale models, have been proposed in order to take reservoir release and storage in GHMs into account (Nazemi and Wheater, 2015). These algorithms can be divided into two groups. First, there are simulation-based algorithms which use functional rules that rely on initial storage, inflows and demand pressure to simulate the release. Secondly, there are optimisation-based algorithms, which try to find the optimal releases to comply with competing water demands using ideal storages at the end of an operational year, initial storages and expected or forecasted inflows and demands.

Hanasaki et al. (2006) proposed a simulation-based scheme that uses the storage capacity, purpose, simulated inflow and downstream water demand of a reservoir. Döll et al. (2009), Biemans et al. (2011) and Voisin et al. (2013) proposed variations on this scheme. The parameters used by these algorithms are easily obtainable. The storage capacity and the main reservoir purpose can be found in databases like GRAND (Lehner et al., 2011), ICOLD (ICOLD, 1998), while inflow and downstream water demand are typically derived by the hydrological model. Although these algorithms perform better than traditional lake

routing algorithms, they remain biased, especially in highly regulated catchments and in cold regions (Biemans et al., 2011; Hanasaki et al., 2008; Pokhrel et al., 2011).

Recently, more data driven simulation-based schemes have been proposed by Wisser et al. (2010b) and Wu and Chen (2011). Both studies propose parametric relationships requiring observed downstream discharges for calibration. Wisser et al. (2010b) use observed data to empirically determine a pair of constants. Wu and Chen (2011), use the Shuffled Complex Evolution (SCE-UA) method (Duan et al., 1992) to optimise several parameters for each individual reservoir, resulting in a better performance than a simple target release scheme, as used in the Soil and Water Assessment Tool (SWAT) (Arnold et al., 1998), or a multi-linear regression algorithm. Unfortunately, the scheme was only tested on a single reservoir and it remains unclear how it performs under different circumstances.

Haddeland et al. (2006b) suggest a retrospective optimisation-based algorithm, whereby knowledge of future inflows is required, that uses the Shuffled Complex Evolution Metropolis (SCEMUA) method (Vrugt et al., 2003) to calculate the optimal release, within a predetermined daily feasible release range, based on the reservoir purpose. Adam et al. (2007) use this algorithm to study the influence of reservoirs on stream-flow in the major Eurasian rivers discharging into the Arctic Ocean after several slight alterations with regards to the determination of the daily allowed release range. Van Beek et al. (2011) further alter the algorithm in order to use it as a prospective model, substituting the future inflows with a function using the inflow in the same month of the previous years. Similar to the simulation-based algorithms, the optimization-based algorithms result in more accurate discharges than traditional lake routing algorithms, but substantial deviations between simulated and observed flows still remain (Adam et al., 2007; Haddeland et al., 2006a).

As a result of limitations of macro-scale algorithms, which are not yet capable of fully mimicking the dynamics of regulated flows, simulations with GHMs are still highly uncertain (Haddeland et al., 2011, 2014). An important opportunity to improve GHMs is by enhancing the simulation-based reservoir operation algorithms (Nazemi and Wheater, 2015).

Hejazi et al. (2008) investigated the role of (uncertainty in) hydrological information in reservoir operation release decisions, realizing that the link between them is human behavior. They find that release decisions strongly rely on the current months inflow, the previous months storage levels and inflow and, to a lesser extend, the predicted inflow for the next month. The simulation and optimisation algorithms tend to neglect human behavior towards uncertainty in hydrological information by assuming that dams are operated in a completely rational way. The proposed method incorporates this aspect in the modelling approach.

Furthermore, the discussed simulation based algorithms use reservoir characteristics from databases like the aforementioned GRAND (Lehner et al., 2011) that contains 6,862 reservoirs. Since more than 40,000 large reservoirs exist today (Takeuchi et al., 2002), the proposed method avoids using databases like GRAND and uses variables that can potentially be observed on a global scale with Earth Observation Satellites, although in this study in-situ observations are used.

Just like the aforementioned data driven, simulation-based schemes, the proposed method requires time-series of observed data to calibrate, or train, the algorithm. Although this training can be computationally expensive, afterwards the simulated releases can be acquired easily. Moreover, the temporal resolution of the proposed method is flexible and dependent on the resolution of the provided time-series.

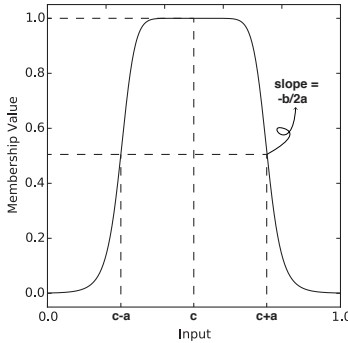

**Figure 3.** A membership-function with a indication of the physical meaning of its parameters.

**Table 1.** Overview of all considered reservoirs, data from Lehner et al. (2011) unless otherwise mentioned.

| Dam Name | Country | Period | Purpose | Inflow [m$^3$/yr]·10$^8$ | Impoundment Ratio[a] [-] | Height [m] | Lon., Lat. [DD] |
|---|---|---|---|---|---|---|---|
| Andijan (AJ) | Uzbekistan | 2001-2010 | Hydropower | 42.0 | 3.97 | 115 | 73.06, 40.77 |
| Bull Lake (BL) | U.S.A | 2001-2013 | Multipurpose[b] | 2.07 | 2.06 | 25[b] | -109.04, 43.21 |
| Canyon Ferry (CF) | U.S.A | 2001-2013 | Multipurpose[b] | 38.1 | 1.95 | 69[b] | -111.73, 46.65 |
| Chardara (CD) | Kazakstan | 2001-2010 | Irrigation | 185 | 5.94 | 29 | 67.96, 41.25 |
| Charvak (CV) | Uzbekistan | 2001-2010 | Hydropower | 70.6 | 5.66 | 168 | 69.97, 41.62 |
| Kayrakkum (KR) | Tajikistan | 2001-2010 | Hydropower | 207 | 7.76 | 32 | 69.82, 40.28 |
| Nurek (NR) | Tajikistan | 2001-2010 | Irrigation | 209 | 2.53 | 300 | 69.35, 38.37 |
| Seminoe (SN) | United States | 1951-2013 | Irrigation | 12.0 | 1.68 | 90 | -106.91, 42.16 |
| Toktogul (TT) | Kyrgyzstan | 2001-2010 | Hydropower | 140 | 1.04 | 215 | 72.65, 41.68 |
| Tuyen Quang (TQ) | Vietnam | 2007-2011 | Hydropower | 97.2 | 7.46 | 92 | 105.40, 22.36 |
| Tyuyamuyun (TM) | Turkmenistan | 2001-2010 | Irrigation[c] | 30.7 | 7.42 | - | 61.40, 41.21 |

[a] The ratio of mean annual inflow to mean annual storage.

[b] U.S. Bureau of Reclamation

[c] Schlüter et al. (2005)

## 3 Methodology

### 3.1 Fuzzy Logic

To model a process, Fuzzy logic uses rules of the form 'IF x is A AND y is B, THEN z is C', where {x,y,z} are linguistic variables, such as storage-level, inflow or release, and {A,B,C} are linguistic values like, 'very high', 'low' or 'very low'. These rules consist of a premise and consequence part and are believed to be able to capture the reasoning of a human working in an environment with uncertainty and imprecision (Shrestha et al., 1996).

Fuzzy reasoning is the process in which fuzzy rules are used to transform input into output and consists of four steps. (1) Firstly, the input variables are fuzzified, (2) next the firing strength of each rule is determined. (3) Thirdly, the consequence of each rule is resolved and (4) finally the consequences are aggregated. In Figure 1, these steps are visualised and in Appendix A an example is given.

A big drawback of fuzzy logic is the need to assess fuzzy rules. Transforming human knowledge or behaviour into a representative set of rules manually is a complicated task. As the amount of input variables and membership functions increases, the total number of required rules quickly becomes very large.

Jang (1993) dealt with this problem by developing a method called Adaptive-Network-based Fuzzy Inference System (ANFIS) to construct a set of fuzzy if-then rules with appropriate membership functions using an Artificial Neural Network (ANN). ANNs are computational models inspired by biological neural networks, that are capable of learning and generalising from examples (Flood and Kartam, 1994). Jang (1993) successfully tested his method on several highly non-linear functions and used it to predict future values of chaotic time-series.

## 3.2 Adaptive-Network-Based Inference Systems

ANFIS is a specific ANN that can deal with linguistic expressions used in fuzzy logic. The network structure is capable of adjusting the shape of the membership functions and of the consequence parameters that form the fuzzy rules, by minimising the difference between output and provided targets. ANFIS is a feed-forward neural network with five layers as seen in Figure 2.

Jang (1993) proposes four training methods in his study, of which one is called the Hybrid Learning Rule (HLR). This method combines gradient descent learning and a least squares estimator (LSE) to update the network parameters. It has an advantage over the other methods because it converges fast and is less likely to become trapped in local minima, which is a common problem when using the gradient descent method. The training consists of two passes which are discussed in more detail below. The network has two parameter sets, the premise and the consequence parameters, situated in the "Membership" and "Implication" layer respectively. The consequence parameters are updated in the forward pass with the LSE, while the premise parameters are updated in the backward pass by gradient descent learning.

### 3.2.1 Forward Pass

In the forward pass, the output of each layer for a given input is calculated and the consequence parameters are adjusted with the LSE, before the final output is generated. Each layer is discussed individually below.

1. The first layer is called the membership layer, the input is put through a membership function to determine its membership value:

$$O_i^1 = \mu_{A_j}(x) \tag{1}$$

where $A_j$ is the $j^{th}$ linguistic label associated with the input type $A$ of $x$. Equation 1 is the membership function of $A_j$, x is the input to the $i^{th}$ node and $\mu$ defines the shape of the membership function (also see Figure 2). Here it is:

$$\mu\left(x\right) = \frac{1}{1 + \left[\left(\frac{x-c_i}{a_i}\right)^2\right]^{b_i}} \tag{2}$$

where $\alpha = \{a_i, \, b_i, \, c_i\}$ are the premise parameters. They determine the shape of the membership function as in Figure 3.

2. The circular nodes in this layer are marked with $\Pi$ in Figure 2. This layer determines the firing strength for all possible combinations of inputs and their associated membership functions:

$$O_i^2 = w_i = \mu_{A_j}\left(x\right) \cdot \mu_{B_k}\left(y\right) \tag{3}$$

where $B_k$ is the $k^{th}$ linguistic label associated with the input type $B$ of $y$.

3. In the third layer, the firing strengths of all nodes are normalized with respect to each other:

$$O_i^3 = \overline{w}_i = \frac{w_i}{\sum_{i=1}^{n} w_i} \tag{4}$$

where $n$ is the total number of fuzzy rules.

4. The fourth layer is called the implication layer. The consequence of each rule is calculated as a linear combination of the input variables, as described by Takagi and Sugeno (1985), and then multiplied by its associated normalized firing strength:

$$O_i^4 = \overline{w}_i \cdot f_i = \overline{w}_i \cdot \left(p_i x + q_i y + r_i\right) \tag{5}$$

in which $\{p_i, \, q_i, \, r_i\}$ are the consequence parameters to be updated by the LSE. Note that the number of consequence parameters increases with the number of input variables.

5. In the fifth layer all the incoming signals are summed to compute the final output:

$$O^5 = \sum_{i=1}^{n} \left(\overline{w}_i \cdot f_i\right) \tag{6}$$

### 3.2.2    Least Squares Estimator

Before the final output is calculated, the consequence parameters need to be updated. The final output can also be written as:

$$\begin{aligned} O^5 = \left(\overline{w}_1 x\right) p_1 + \left(\overline{w}_1 y\right) q_1 + \left(\overline{w}_1\right) r_1 + \ldots \\ + \left(\overline{w}_n x\right) p_n + \left(\overline{w}_n y\right) q_n + \left(\overline{w}_n\right) r_n \end{aligned} \tag{7}$$

If P combinations of input and target values, or P samples, are provided for training the network, the output for all inputs is given by:

$$\begin{bmatrix} O_1^5 \\ \vdots \\ O_P^5 \end{bmatrix} = A \cdot X \tag{8}$$

In which the dimensions of A and X are respectively $(P \cdot M)$ and $(M \cdot 1)$, with M indicating the total number of consequence parameters.

Equation 8 needs to be equal to the target values, B, provided by each sample:

$$A \cdot X = B \tag{9}$$

This is an overdetermined problem which generally does not have a exact solution. Therefore, a least square estimate is sought with sequential formulas (Aström and Wittenmark, 2011):

$$X_{i+1} = X_i + S_{i+1} \cdot a_{i+1} \cdot \left(b_{i+1}^T - a_{i+1}^T \cdot X_i\right)$$

$$\tag{10}$$

$$S_{i+1} = S_i - \frac{S_i \cdot a_{i+1} \cdot a_{i+1}^T \cdot S_i}{1 + a_{i+1}^T \cdot S_i \cdot a_{i+1}}$$

with,

$$i = 0, 1, \cdots, P - 1$$

$X_0 = 0$;

$S_0 = \gamma \cdot I$;

$\gamma$ = positive large number

$I$ = identity matrix with dimension $(M \cdot M)$

$a_i^T = i^{th}$ row vector of matrix A;

$b_i^T = i^{th}$ element of B;

So during every forward pass, the consequence parameters, X, are updated. Note that for one update, only one row of matrix A and only one target value is needed. One sample results in one update of the consequence parameters. After the parameters of layer 4 have been updated with equation 10, equation 6 is used to calculate the output. Finally, the error rate can be calculated
with:

$$E_p = (T_p - O_p^5)^2 \tag{11}$$

in which $T_p$ is the target value and $O_p$ the output value for the $p^{th}$ sample. After the error rate has been determined, the forward pass is finished and the error rate is propagated back through the network in order to update the premise parameters with the gradient descent method.

**3.2.3  Backward Pass**

During the backward pass, the error associated with the sample under consideration is propagated backward through the network in order to acquire the gradient of the error with respect to each individual premise parameter. So, $\alpha$ is updated according to:

$$\Delta \alpha = -\eta \cdot \frac{\partial E_p}{\partial \alpha} \tag{12}$$

In which $\eta$ is the learning rate, which is defined as:

$$\eta = \frac{k}{\sqrt{\sum_\alpha \frac{\partial E_p}{\partial \alpha}^2}} \tag{13}$$

where $k$ is the step-size that determines the speed of convergence. The value of $k$ is chosen and changed heuristically. When the error measure decreases for four consecutive steps, the step-size increases by 5%. After the occurrence of two consecutive oscillations of the error measure, the step-size decreases by 5%.

The derivative in Equation 12 and 13 is defined as:

$$\frac{\partial E_p}{\partial \alpha} = \frac{\partial E_p}{\partial O^5} \frac{\partial O^5}{\partial O^4} \frac{\partial O^4}{\partial O^3} \frac{\partial O^3}{\partial O^2} \frac{\partial O^2}{\partial O^1} \frac{\partial O^1}{\partial \alpha} \tag{14}$$

The first term on the right side of equation 14 can be derived from equation 11:

$$\frac{\partial E_p}{\partial O^5} = -2(T_p - O_p^5) \tag{15}$$

The final term of equation 16 is derived from Equation 4 as:

$$\frac{\partial O^1}{\partial \alpha}
\begin{cases}
\dfrac{\partial O^1}{\partial a} = \dfrac{\left( 2\,b\,(-c+x)^2 \left( \frac{(-c+x)^2}{a^2} \right)^{(-1+b)} \right)}{\left( a^3 \left( 1 + \left( \frac{(-c+x)^2}{a^2} \right)^b \right)^2 \right)} \\[3em]
\dfrac{\partial O^1}{\partial b} = -\left( \dfrac{\left( \left( \frac{(-c+x)^2}{a^2} \right)^b Log\left[ \frac{(-c+x)^2}{a^2} \right] \right)}{\left( 1 + \left( \frac{(-c+x)^2}{a^2} \right)^b \right)^2} \right) \\[3em]
\dfrac{\partial O^1}{\partial c} = \dfrac{\left( 2\,b\,(-c+x) \left( \frac{(-c+x)^2}{a^2} \right)^{(-1+b)} \right)}{\left( a^2 \left( 1 + \left( \frac{(-c+x)^2}{a^2} \right)^b \right)^2 \right)}
\end{cases} \tag{16}$$

The other terms in Equation 14 can easily be derived from Equations 3-6.

After the update of the premise parameters, a next sample is provided to the network and the forward pass starts again. When all samples have been passed trough the network once, one epoch has passed and another epoch is started until the solution converges.

In summary, first the input part of a sample is used to activate the network and, together with the target of the same sample, the consequence parameters are updated using a LSE. Next, the output error is calculated with Equation 11 and propagated backwards through to network with Equation 14, after which Equation 12 is used to adjust the premise parameters. Once the backward pass has been completed, the next sample is used to start again, until the error rate converges.

## 3.3 Data

In order to determine whether ANFIS is capable of deriving a set of useful fuzzy rules that captures the characteristics of how a dam is operated, 11 reservoirs for which in-situ measurements were readily available have been investigated. Table 1 lists the considered dams, which are located in the United States, Vietnam and several Central Asian countries, together with

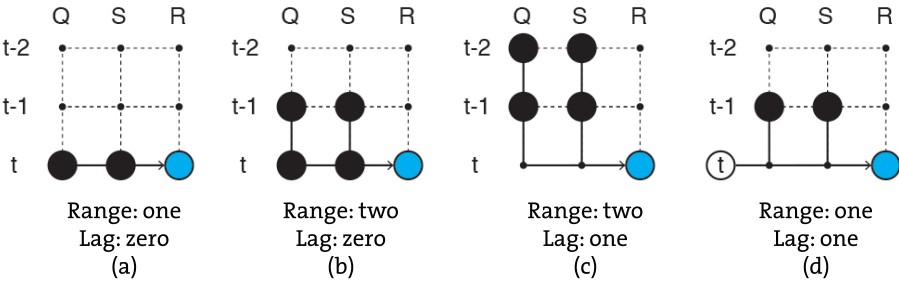

**Figure 4.** Diagrams showing different sample set-ups, The black dots represent input parameters, while the blue dot shows the target.

their respective purpose, mean annual inflow, ratio of mean annual inflow to mean annual storage (impoundment ratio), dam height, location and the period over which data on inflow, storage and release is available. The size of the dams varies with dam heights ranging between 25 and 300 meters. The purpose of the reservoirs is also diverse, several hydro-power, irrigation and two multi-purpose reservoirs are considered. The periods of available data are around 10 years for most dams. For Tuyen Quang there is a significantly shorter period of available data (5 years) and for Seminoe dam in the United States there is 62 years of available data. The data of the Central Asian reservoirs has been converted from a 10-day to a monthly time-scale, while the data-series of reservoirs in the United States and Vietnam have been converted from daily to monthly data. This has been done in order to allow comparison between all reservoirs.

### 3.4 Settings

To train a network, the first 60% of the dataset of each dam is used to train the parameters, the next 20% is used to validate the solution . Finally, the remaining 20% is used to test the solution. During an epoch, all samples in a training-set are passed forward and backward through the network once. The training is stopped when for at least five consecutive epochs, the mean square error (MSE) of the simulation with respect to the validation-set has increased, after which the configuration of the network with the lowest validation MSE is chosen.

At this point, the training-set has been used to update the networks parameters and the validation-set has been used to select the state of the network for which the results matched best with data not present in the training-set. Since the validation-set has been used to select the best configuration of the network, a third and independent set is used to test the performance of the network. This third set is the test-set.

Initially, two variables will be used as input to train the network, storage (S) and inflow (Q), while the release (R) will be used as a target or output of the network. A simple configuration of the network could be formulated as:

$$Input = \{S(t)[2], Q(t)[2]\},$$

$$\tag{17}$$

$$Target = \{R(t)\}$$

This sample type has a prediction horizon of zero time-steps, the output of the network will be the release of a reservoir for the same month as the input provided. The time-range of this sample is one, because the input parameters are considered at time $t$ only. Figure 4a shows this sample type in a schematic way. The numbers within square brackets indicate how many membership functions are used for the particular variable.

A somewhat more complicated sample is the following:

$$Input = \{S(t)[2], S(t-1)[2], Q(t)[2], Q(t-1)[2]\},$$

(18)

$$Target = \{R(t)\}$$

Which has a time-range of two and also no prediction horizon, see Figure 4b. With this setup, the release at time $t$ is determined using the storage and inflow at time $t$ and $t-1$. Note that since there are now four input variables, the complexity of the network increases. Two membership functions are used per input parameter, so 8 membership functions are needed in total.

With three variables per membership-function, see Equation 1, the membership layer contains 24 parameters. Furthermore, $2^4 = 16$ different rules can be created with this input. Since the consequence of every rule contains as many parameters as the length of the input array plus one, see Equation 5, the implication layer will contain $5 \cdot 16 = 80$ parameters. By varying the time-range, prediction horizon and the number of membership functions used per input parameter, it is possible to generate many different sample configurations. Increasing the prediction horizon of Equation 18 results in the following sample set-up:

$$Input = \{S(t-1)[2], S(t-2)[2], Q(t-1)[2], Q(t-2)[2]\},$$

15                                                                                                          (19)

$$Target = \{R(t)\}$$

With this set-up the release is predicted one time-step ahead of the input variables, also see Figure 4c.

Additionally, since seasonality plays an important role in the operation of reservoirs, a third input parameter will also be considered, the Time-of-the-Year (ToY). For example, like this:

$$Input = \{S(t-1)[2], Q(t-1)[2], ToY(t)[2]\},$$

(20)

$$Target = \{R(t)\}$$

Figure 4d shows an example of a sample using the ToY. Since the ToY is used with two membership functions, as indicated between the square brackets, it can be thought of as a parameter indicating whether the season is either "dry" or "wet".

Finally, in order to use back-propagation, initial values for the parameters of the membership layer need to be set. These are set such that for any input, the sum of the membership functions equals one, an example for an input parameter with two membership functions can be seen in Figure 5.

**3.5   Comparison with a macro-scale reservoir algorithm**

In order to compare simulated releases with those made by an existing macro-scale algorithm, the data used to train the networks has also been applied to the algorithm proposed by Hanasaki et al. (2006) (from here on referred to as HNS). This

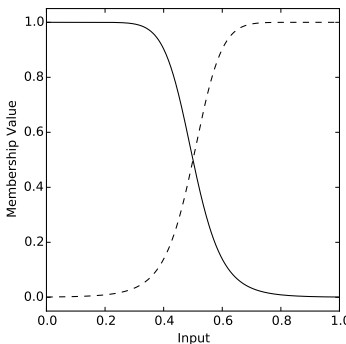

**Figure 5.** Example showing the initial membership functions for a variable consisting of two membership functions

algorithm makes a distinction between irrigation and non-irrigation reservoirs. For irrigation reservoirs, the algorithm requires data on water demands. Since the method proposed in this study does not require water demands, the irrigation reservoirs (Chardara, Nurek Seminoe and Tyuyamuyun) have been omitted from this comparison.

The monthly release for the remaining reservoirs is calculated as:

$$r_{m,y} = \begin{cases} k_{rls,y} \cdot r'_{m,y} & (c \geqslant 0.5) \\ \left(\frac{c}{0.5}\right)^2 \cdot k_{rls,y} \cdot r'_{m,y} + \left(1 - \left(\frac{c}{0.5}\right)^2\right) \cdot i_{m,y} & (0 \leqslant c < 0.5) \end{cases} \tag{21}$$

Where $c$ is the storage capacity divided by the mean total annual inflow, $r'_{m,y}$ is the provisional monthly release, which equals the mean annual inflow for non-irrigation reservoirs, $i_{m,y}$ is the current months inflow and $k_{rls,y}$ is the release coefficient, defined as:

$$k_{rls,y} = \frac{S_{first,y}}{\alpha \cdot C} \tag{22}$$

In which $S_{first,y}$ is the storage at the beginning of an operational year, $\alpha$ is a dimensionless constant set to 0.85 and $C$ is the total storage capacity of the reservoir.

To prevent reservoirs from overflowing, excess storage left after water for the current month has been released is released additionally.

## 4  Results

### 4.1  Simple Set-Up

Simulating reservoir releases with a simple set-up as in Equation 17 results in MSEs ranging from $5.80{\cdot}10^{-3}$ to $41.1{\cdot}10^{-3}$ and Nash-Sutcliffe (NS) coefficients from 0.33 to 0.95, ignoring the outlier Chardara with an MSE of 71.2 and NS coefficient of -0.49, see Table 2. Compared to HNS, five out of the seven non-irrigation reservoirs score better on one or both of the indicators.

**Table 2.** The test MSEs ($10^{-3}$) [-] and the NS coefficients [-] for all dams for different time-ranges and with different prediction horizons together with the indicators using the Hanasaki et al. (2006) (HNS) method. Because HNS requires additional data for irrigation reservoirs, CD, NR, SN and TM have been omitted. Bold numbers indicate indicators with better performance than HNS.

| Range | Lag | | AJ | BL | CF | CD | CV | KR | NR | SN | TT | TQ | TM |
|-------|-----|-----|------|------|------|------|------|------|------|------|------|------|------|
| | | | | | | | Dam | | | | | | |
| 1 | 0 | MSE | 23.9 | **41.1** | **5.80** | 71.2 | **5.68** | 23.6 | 15.2 | 16.0 | **21.1** | 12.3 | 19.8 |
| | | NS | **0.69** | **0.46** | **0.80** | -0.49 | **0.92** | 0.45 | 0.78 | 0.40 | **0.33** | 0.50 | 0.95 |
| 2 | 0 | MSE | **5.10** | **15.8** | **1.85** | 4.13 | 32.3 | **6.27** | 3.31 | 11.6 | **9.60** | **6.18** | 0.981 |
| | | NS | **0.93** | **0.79** | **0.94** | 0.91 | 0.54 | **0.85** | 0.95 | 0.57 | **0.70** | 0.75 | 0.98 |
| 2 | 1 | MSE | 41.0 | **31.9** | **5.78** | 23.6 | **13.0** | 32.6 | 23.0 | 12.0 | **28.0** | 24.1 | 21.5 |
| | | NS | 0.46 | **0.58** | **0.80** | 0.51 | **0.81** | 0.23 | 0.66 | 0.55 | **0.12** | 0.01 | 0.5 |
| 2 | 2 | MSE | 46.6 | **41.5** | 21.5 | 48.3 | 30.7 | 115 | 40.2 | 21.9 | 39.1 | 50.8 | 34.6 |
| | | NS | 0.42 | **0.45** | **0.24** | -0.02 | 0.55 | -1.67 | 0.39 | 0.18 | -0.19 | -0.91 | 0.21 |
| HNS | | MSE | 21.9 | 48.9 | 6.34 | - | 13.2 | 15.2 | - | - | 28.6 | 7.57 | - |
| | | NS | 0.51 | 0.11 | 0.22 | - | 0.70 | 0.52 | - | - | 0.02 | 0.83 | - |

Because the membership-functions of Andijan and Charvak show different effects the training can have on the membership functions and their convergence curves show two extremes (very fast and very slow convergence respectively), they are presented more in-depth below. The inputs, Q and S, for both reservoirs vary significantly over the years.

For Andijan, the validation-set contains two very dry years with low inflows and low storage-levels, while the peak flows in the rest of the dataset are of similar magnitude, see Figure 6a. Consequently, the observed releases, $R_{obs}$, in the two dry years are also relatively low, see Figure 6b.

The storage-level of Charvak reservoir reaches its maximum nearly every year, while the inflow during several years is not more than 50% of the inflow during wetter years. Nevertheless, even during some of these drier years, it appears the reservoir is able to fill completely, see Figure 7.

$R_{sim}$ follows the test data for both reservoirs with MSEs of $23.9 \cdot 10^{-3}$ and $5.68 \cdot 10^{-3}$ and NS coefficients of 0.69 and 0.92 for Andijan and Charvak respectively, as can be seen in the first two rows of Table 2. Most of the peaks in the test-set match closely, only the first peak in the Andijan test-set is too low.

The shape of the four membership functions of Andijan differ from their initial shapes, see Figure 8a and b. The membership-function for low inflow changed the least, while the high inflow function has shifted to the left, see the initial shapes in Figure 5, intersecting each other around an inflow of 0.4. Both membership functions for storage have shifted to the right, intersecting each other around an input of 0.6. When the storage is larger than 0.6, a different consequence rule will be used to calculate the release. This network configuration, resulting in the lowest validation-error, was reached after two epochs, see Figure 8c.

The membership functions of Charvak for reservoir inflow have moved slightly to the left and the steepness of the bell shapes has increased for the low inflow membership-function and decreased for the other. There is a clear distinction between

consequences for inflows below and above 0.4, see Figure 9a. The membership functions for storage have moved away from each other. Storages between 0.4 and 0.6 now result in the activation of two rules with approximately similar firing strengths. The release for situations with storages between these values will be aggregated from two fuzzy rules, see Figure 9b. The training of the network for Charvak takes a lot longer than for Andijan, with more than 200 epochs, although the difference in
error is minimal as seen in Figure 9c.

The membership functions for other reservoirs have a similar shape as for Andijan and Charvak. Occasionally, multiple membership functions dominate over the same part of the input domain, resulting in the simultaneous activation of fuzzy rules. Sometimes both membership functions become near zero for a part of the domain, like the storage membership functions of Charvak, resulting in simultaneous activation of two rules. The rule for low inflow and storage is most frequently activated
for the majority of reservoirs, followed by the rule for a low inflow and a high storage. The rules with regards to high inflows are used less frequently, see Figure 10a. The simulation of Kayrakkum is done using only the rule for low inflow and high storage, implying that the high inflow and the low storage membership functions are zero over their entire domains. $R_{sim}$ for Kayrakkum is solely based on one consequence rule, as in Equation 5.

The consequence parameters of rules associated with a low inflow and storage, and a low inflow and high storage are quite
similar across the different reservoirs, see Figure 11. For example, the rule for a "low" inflow and a "low" storage for most reservoirs consists of the weighted sum of the two input parameters, $Q \cdot p_1 + S \cdot q_1$, added to the third independent variable $r_1$, where $p_1$, $q_1$ and $r_1$ have values around respectively 0.45, 0.10 and 0.05. The range of consequence parameters of the remaining two rules is larger, the consequences of these rules differ more per reservoir. Most of the outliers belong to three reservoirs (Chardara, Toktogul and Bull Lake).

The test-set for $R_{sim}$ and $R$ for the other nine reservoirs are shown in Figure 12. Of the 11 tested reservoirs, Chardara is the worst performing, see Table 2 and Figure 12c. Although the shape of $R_{sim}$ somewhat resembles the observed values, the high and low flows occur at the correct time, the values are far off. The trained network of Chardara utilizes all its rules, see Figure 10a, but this is either not sufficient to capture the operational modes of the reservoir or the validation and test-sets differ significantly from the training-set.

The MSEs and NS coefficients for Bull Lake and Kayrakkum are better than those of Chardara, see Table 2. Although the peak releases in $R_{sim}$ for Bull Lake are similar to the observed ones, the low flows are not very accurate. The model is not able to deal with the near zero flows during the dry season, see Figure 12a. The simulated releases for Kayrakkum are of the right magnitude as can be seen in Figure 12d, only during the first year of the test-set, the annual release has been lower than usual and the model appears unable to cope with this phenomenon. This low annual flow was not present in the training data-set,
explaining why the model does not use more of its available parameters.

$R_{sim}$ for Togtogul, Tuyen Quang, Nurek and Canyon Ferry clearly follows $R_{obs}$, the magnitude and timing of low and peak flows match, see Figure 12g,h,e and b. For Tuyen Quang, it is important to note once more that the dataset is very short and the test-set is only 10 months long.

Seminoe has the largest dataset and shows a similar problem as Bull Lake. The network seems incapable of dealing with the
very low flows and the high peak-flows, while the medium peaks are simulated quite accurately, see Figure 12f.

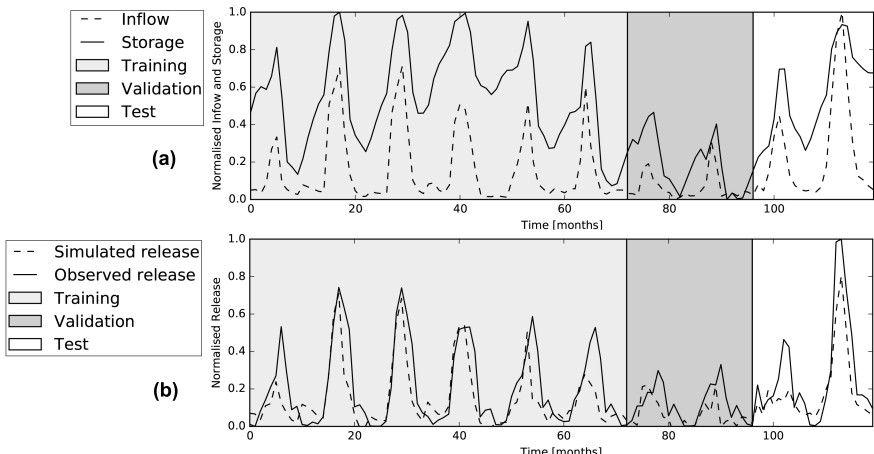

**Figure 6.** Graphs showing the (a) inflow and storages and the (b) simulated and observed releases for Andijan reservoir for the training, validation and test-set.

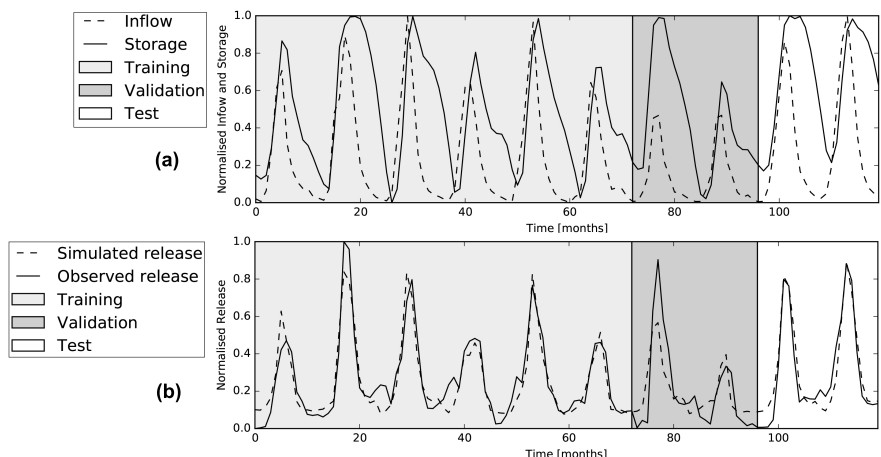

**Figure 7.** Graphs showing the (a) inflow and storages and the (b) simulated and observed releases for charvak reservoir for the training, validation and test-set.

Finally, Tyuyamuyun performs very well, with a very accurate timing and magnitude of peak and low flows, see Figure 12i. This result can be explained by comparing $R_{obs}$ with the inflow, which shows a very strong linear correlation.

## 4.2 Additional Variables

The MSEs for the networks of the 11 reservoirs trained with a sample set-up as in Equation 18 range between $0.981 \cdot 10^{-3}$ and $32.3 \cdot 10^{-3}$ and the NS coefficients between 0.54 and 0.98, see the third and fourth row in Table 2. Comparison of the errors with the errors of the simpler set-up, like Equation 17, shows clearly that the performance of the ANN improves. This also becomes

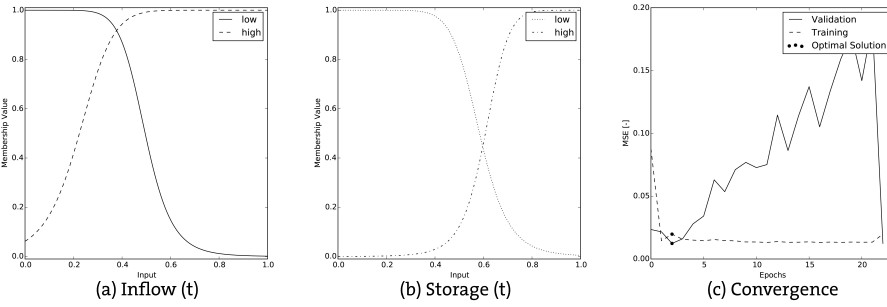

(a) Inflow (t)    (b) Storage (t)    (c) Convergence

**Figure 8.** Results of Andijan Dan. (a) and (b) show the membership functions of the inflow and storage, respectively, after the network has been trained. (c) shows the change of the MSE with respect to the training and validation set.

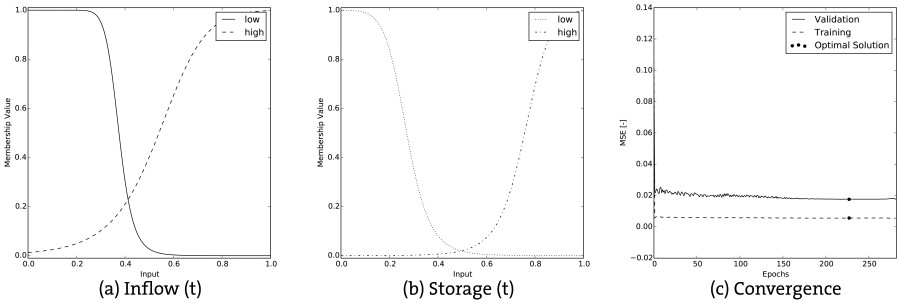

(a) Inflow (t)    (b) Storage (t)    (c) Convergence

**Figure 9.** Results of Charvak Dam. (a) and (b) show the membership functions of the inflow and storage, respectively, after the network has been trained. (c) shows the change of the MSE with respect to the training and validation set.

clear from the dashed lines in Figure 12, which shows $R_{sim}$ for nine reservoirs together with $R_{obs}$ and $R_{sim}$ with a simple set-up. For all 9 reservoirs, the peak and low flows match closely. Consequently, the advantage in performance compared to HNS further increases for most reservoirs.

By using a time-range of two and no prediction horizon, as in Equation 18, 16 rules are available in a network. Surprisingly, many trained networks do not use more than two rules, see Figure 10b. Only Canyon Ferry, Charvak and Seminoe use more than two rules, namely 4, 8 and 13 rules respectively. Apparently, solely the increase in the number of consequence parameters for each rule is sufficient to improve results. Only Seminoe, which uses the longest time-series, appears to really need more rules to describe different situations.

Adding more membership functions or input variables to the configuration of the network increases the number of fuzzy rules. It is clear that increasing the time-range over which Q and S are considered improves results. Comparison of the average test MSEs of the 11 reservoirs for different sample set-ups shows clearly that simply adding more input variables does not always lead to better results. The results are worst when only reservoir storage is used as input, see the bottom row in Figure 13a, with average MSEs around 0.045. When only inflow is used as input, the results are better, with average MSEs around 0.025, see the leftmost column in figure 13a. By using combinations of storage and inflow the average MSE can further

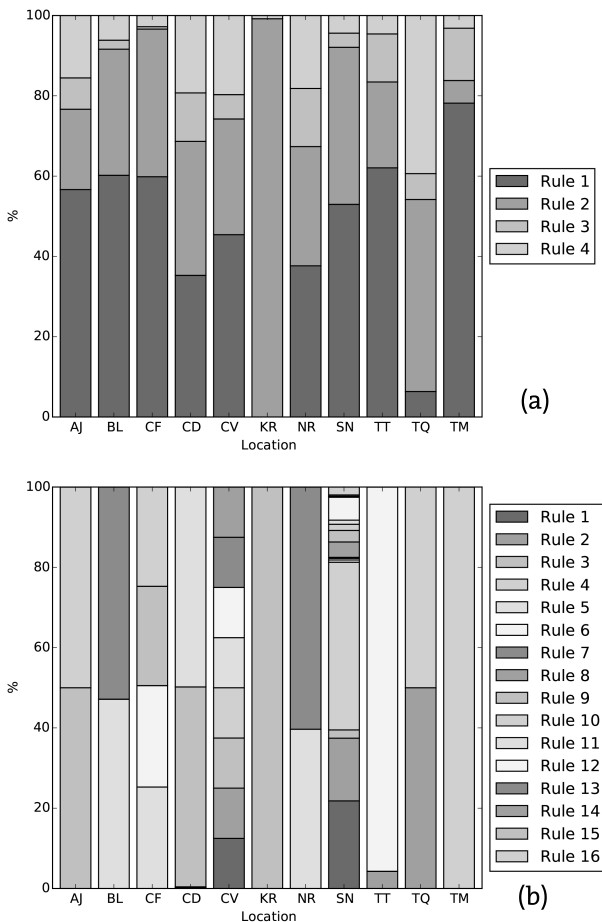

**Figure 10.** Graph indicating how many of the rules available to a network are used for (a) a network with a simple, 4-rule, set-up and (b) a network with a more complex, 16-rule, set-up.

decrease, the simple sample set-up as in Equation 17 however, does not result in a lower average MSE compared to a sample set using solely {Q(t)[2]} as input. Adding an input variable considering the storage at time $t-1$ (input = {Q(t)[2] & S(t)[2] & S(t-1)[2]}) does decrease the average MSE to 0.005, see the second row from the bottom in Figure 13a. This is roughly the same result as achieved by using the sample set-up as in Equation 18. The magnitude of the average MSE for sample set-ups

5   including the ToY is similar to set-ups not using it, see Figure 13b.

Figure 14a presents the significance of adding more input parameters or membership functions to the network. Starting in the bottom left corner, the results for all reservoirs with a simple set-up are compared to a slightly more complex set-up, as indicated by the arrows, using a one-sided student t-test. For example, the set-up using {Q(t)[2] & Q(t-1)[2] & S(-)[-]} for input, the current and previous inflow with two membership functions each and no storage, is compared to the set-up using

10   {Q(t)[3] & S(-)[-]}. The significance of increasing the time-range of the inflow has a one-tailed p-value smaller than 0.10 but

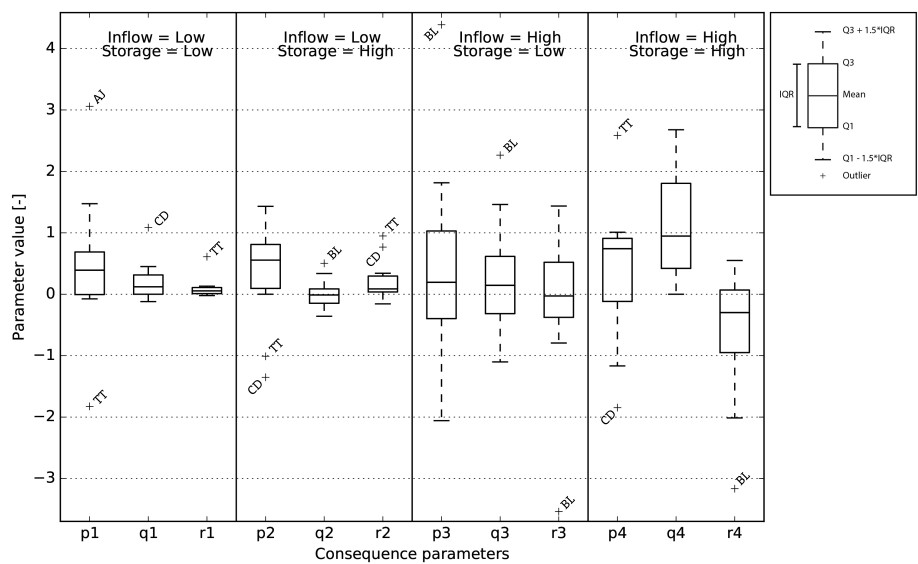

**Figure 11.** The consequence parameters of all reservoirs, separated per rule in a boxplot. The parameter 'p' is multiplied with the inflow, 'q' with storage after which they are summed with 'r' to determine the release. The outliers are labelled as AJ for Andijan, TT for Toktogul, CD for Chardara and BL for Bull Lake.

larger than 0.05. From Figure 14a, it becomes clear that increasing the complexity with the use of storage data leads to better results than adding more complexity with inflow data.

Like Figure 14a, Figure 14b shows p-values indicating the significance of adding more complexity to the network. However, now the addition of the ToY parameter is tested. Each value in Figure 14b shows a comparison between a set-up using the
5  ToY and the same set-up without the ToY parameter, making arrows unnecessary. For example, the set-up using {ToY(t)[2] & Q(t)[2] & S(t)[2]} as input is compared to a set-up using {Q(t)[2] & S(t)[2]} as input. The significance of this addition to the network has a p-value between 0.05 and 0.10. No clear pattern is visible here, it seems like the addition of ToY increases the networks accuracy simply by the increased complexity of the network.

In Figure 15, a similar approach is used. Here the reservoirs have been split in two groups using their impoundment ratios,
10  see Table 1. One group contains reservoirs with impoundment ratio larger than the median (Figure 15a), while the other group contains reservoirs for which the ratio is smaller than the median (Figure 15b). Adding information about storage to the network is clearly more significant for reservoirs with a small impoundment ratio.

## 4.3 Adding a prediction horizon

When adding a prediction horizon of one month to the network, the MSEs range between $12.0 \cdot 10^{-3}$ (Seminoe) and $41.0 \cdot 10^{-3}$
15  (Andijan). For two months the MSEs vary between $21.5 \cdot 10^{-3}$ (Canyon Ferry) and $115 \cdot 10^{-3}$ (Kayrakkum). The NS coefficients

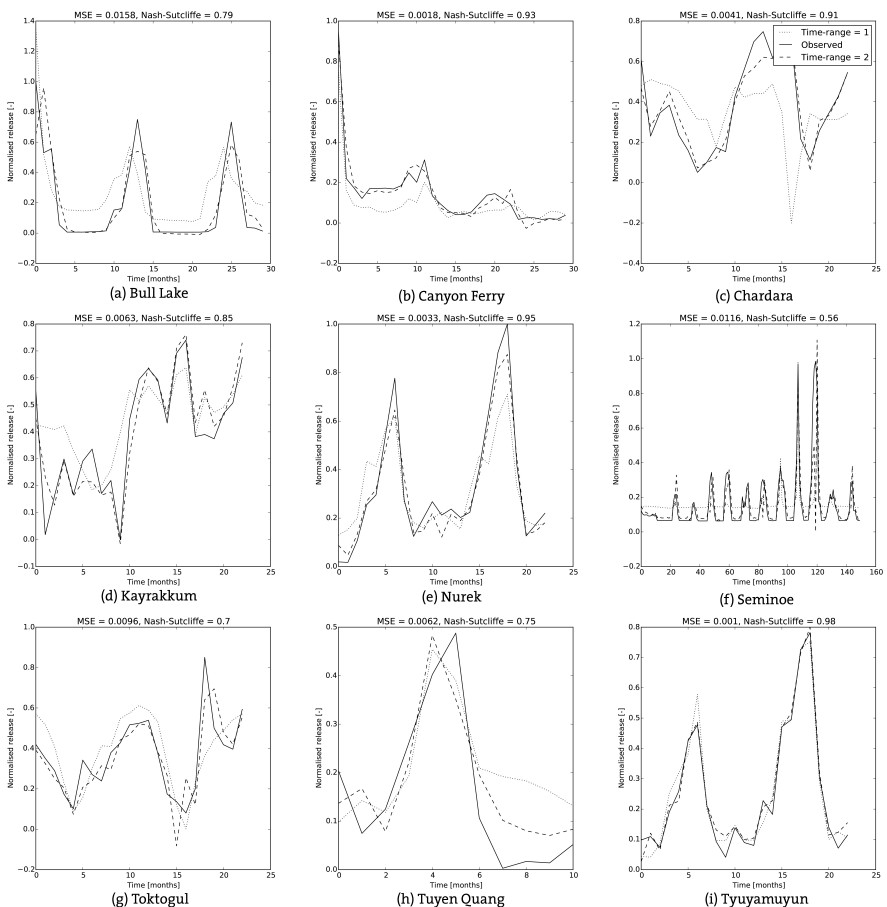

**Figure 12.** Simulated and observed reservoir releases for nine reservoirs when simulated with a time-range of one or two.

range between 0.01 (Tuyen Quang) and 0.81 (Charvak) for a prediction horizon of one month and between -1.67 (Kayrakkum) and 0.55 (Charvak) for a two month prediction horizon (see the last four rows of Table 2). As expected, the overall results worsen as the prediction horizon is increased, although still several reservoirs exhibit better performance than HNS.

## 5 Discussion

### 5.1 Using a simple set-up

A simple configuration of ANFIS, with a time-range of one and no prediction horizon, is capable of determining fuzzy rules that are able to describe the release regime for most reservoirs with MSEs as low as $5.08 \cdot 10^{-3}$, see Table 2. For Bull Lake and Seminoe however, this amount of complexity seems to be insufficient. During the periods of very low flows, the release from these reservoirs is consequently overestimated, see Figure 12a and f. In both cases, all four rules are utilised, see Figure

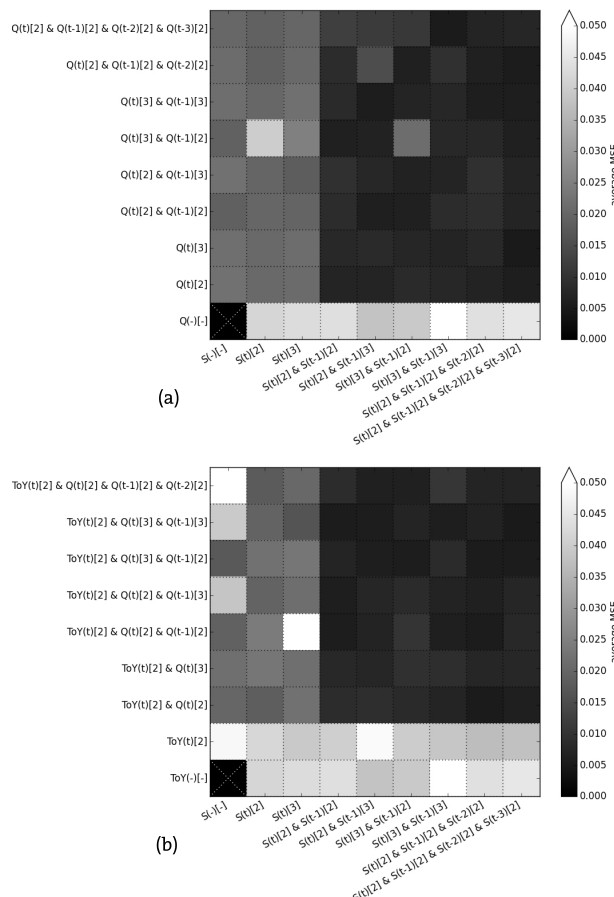

**Figure 13.** Matrix showing the average test MSEs of the 11 considered reservoirs as the number of input variables and membership functions increases. (a) shows combinations of storage and inflow input variables and (b) also includes the Time-of-Year variable.

10a, suggesting that a more complex network is needed. For Seminoe, it is important to note that the length of the dataset is 62 years, a period over which it is not unlikely that the operation regulations might have changed. This would mean the fuzzy rules are trying to describe two different modes of operation.

The classifications made by the membership functions differ per reservoir. These differences can be explained by reservoir characteristics, such as maximum storage capacity, dead storage capacity, impoundment ratio or reservoir purpose. For example, a filling level of 60% at the end of a dry season in a reservoir used for irrigation, will be interpreted differently from a similar filling level in a reservoir mainly used for hydropower.

Besides the variety of physical properties of reservoirs causing differences in how input parameters are classified, two phenomena that are intrinsic to ANFIS seem to be especially relevant. As membership functions move either left or right, it is possible that a membership-function becomes zero in the entire domain, rendering its associated rules obsolete. That is, of the four rules incorporated in the network, only two were left to be used. When this occurs for all input variables, only one rule

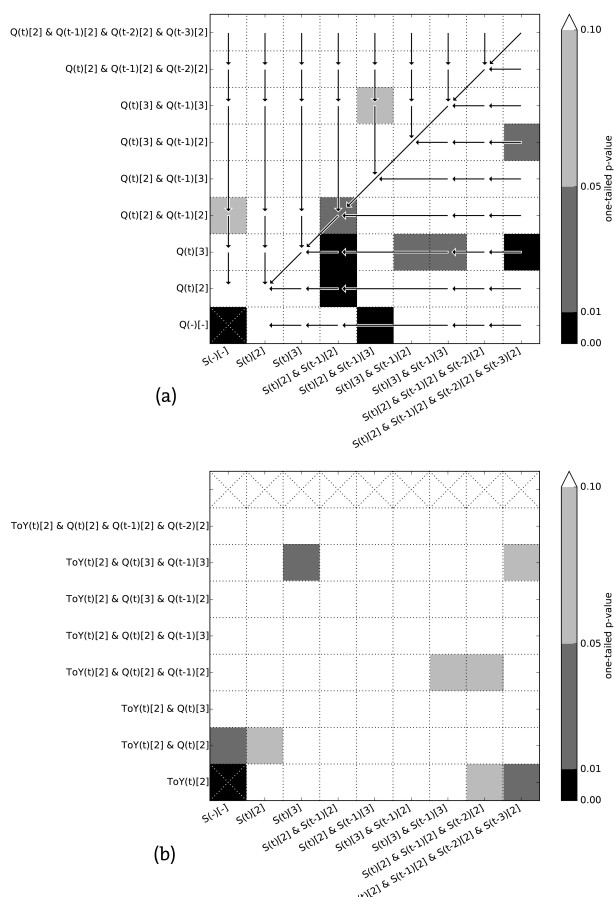

**Figure 14.** Matrix showing the significance (one-sided student t-test) of increasing the complexity of the ANN by adding either more input variables or membership functions. (a) compares sample set-ups with less complex set-ups indicated by an arrow and (b) compares cases with and without Time-of-Year as an input variable.

is left to be used, as is the case for Kayrakkum, see Figure 10. Considering this phenomenon from a physical point of view, one could argue that when this happens, there is no need to make a distinction between two different classifications of an input parameter. Apparently the system under consideration can be described using fewer rules than available.

Secondly, the opposite can happen too. Instead of a membership function moving away from the domain and giving hegemony to the other membership function, two membership functions can also move towards each other. When either the centers of the membership functions, defined by $c$, approach each other or the widths of the peaks, defined by $b$, of the membership functions increase, a large part or the whole domain can become dominated by two membership functions simultaneously. This results in the activation of two fuzzy rules for a single input, which is undesirable because it is illogical and it undermines the interpretability of outcomes.

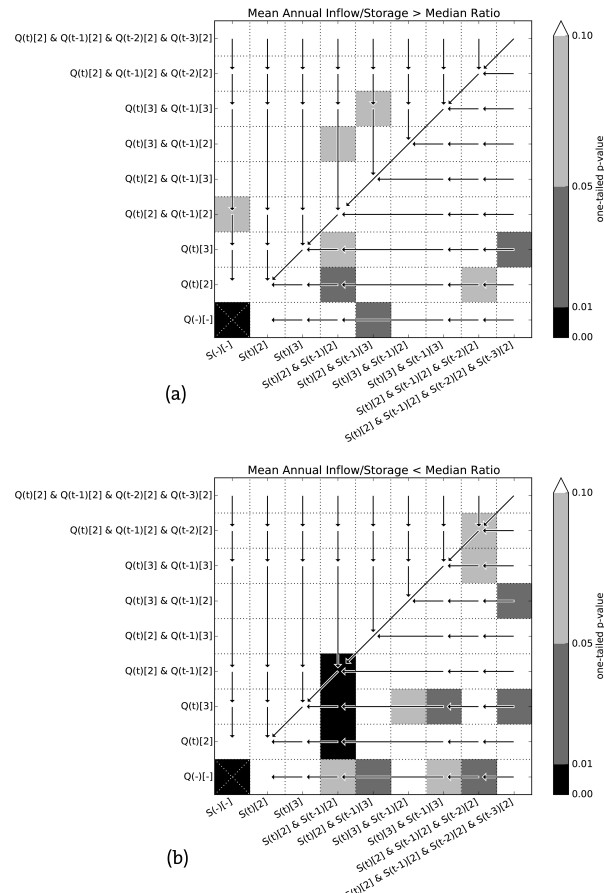

**Figure 15.** Matrix showing the significance (one-sided student t-test) of increasing the complexity of the ANN by adding either more input variables or membership functions for (a) reservoirs with a large impoundment ratio and (b) reservoirs with a small impoundment ratio.

With simple set-ups resulting in a network with four fuzzy rules, these two phenomena occur very infrequently, in most cases all four available rules are used, see Figure 10a.

The range of the consequence parameters, see Equation 5, in the implication layer for all reservoirs ranges from -3 to 3, see Figure 11, although the majority of the parameters lies between -1 and 1. This large range implies that the consequence parts of 5 the fuzzy rules differ a lot for the 11 reservoirs. The consequences associated with 'low' inflows are more similar. Apparently the operating policies of the different reservoirs differ more from each other when the inflow into the reservoir is high. The difference in consequences is not surprising however, since the purposes, sizes, impoundment ratios and associated climates differ greatly among the reservoirs. If a group of very similar reservoirs were considered, the range of these parameters is expected to decrease and perhaps a more general pattern in consequences for a specific type of reservoir could be observed.

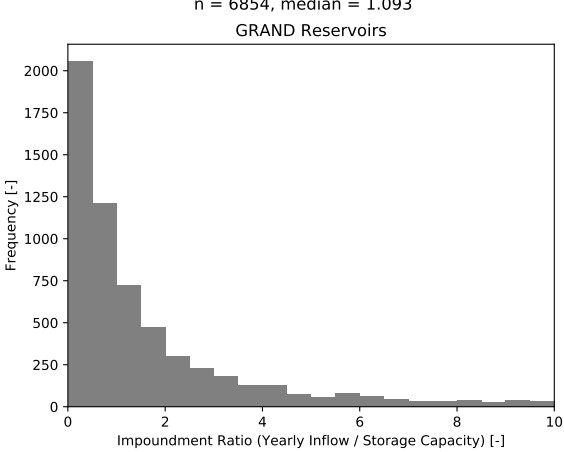

**Figure 16.** The impoundment ratios, defined as the yearly inflow divided by the total storage capacity, of the reservoirs in the GRAND (Lehner et al., 2011) reservoir database.

## 5.2 Increasing complexity

When the complexity of the network is increased, it appears that the aforementioned phenomena of membership functions turning either zero or one over the entire input domain, occur more often. A network trained with a sample set-up as in Equation 18 can utilize up to 16 rules. The output of these networks is generated with a very limited number of rules, see

Figure 10b, generally less than four. Nevertheless, the simulated releases from these networks perform significantly better than their less complex counterparts, see Table 2.

The explanation for this increase in performance regardless of the decrease in rules used is twofold. The most obvious cause lies in the formulation of the consequence of a fuzzy rule, see Equation 5. As the number of input parameters grows, the number of trainable parameters in the implication layer also increases.

Additionally, there is simply more information available. Whereas a four rule network in this study can determine the release from a reservoir based on the current storage and inflow, more complex networks can also consider the storages and inflows further back in time. Figure 14a shows the significance of increasing the complexity of a network and the addition of more information. An important conclusion that can be drawn from the patterns in Figure 14a is that the addition of information about reservoir storage in the previous month is more significant, $p < 0.01$, than the addition of information about the inflow in

the previous month, $0.05 < p < 0.1$. Furthermore, the addition of information on storage even further back in time still improves the results, $p < 0.1$, whereas the inflow this far back in time does not have a significant influence on performance anymore, $p > 0.1$.

This greater value of storage information can be explained by considering the reservoirs mean annual inflow divided by the storage capacity, the impoundment ratio. With a value of 1.04, Toktogul reservoir has the lowest impoundment ratio of the 11

reservoirs, see Table 1. When this ratio is smaller than 1, the storage capacity is larger than the mean yearly inflow. In that case,

the release of the reservoir is unlikely to be very dependent on the current inflow, since the reservoir has a strong buffering capacity. On the other hand, when the impoundment ratio is very large, the mean annual inflow is greater than the storage capacity and the release will approach the inflow.

The 11 reservoirs all have ratios greater than 1, with an average of 4.3. By splitting the considered reservoirs into two groups of equal size, using the median of the 11 impoundment ratios (i.e. 3.97), and testing the significance of increasing the complexity and addition of more information to the network again for both groups, this can indeed be observed, see Figure 15. The performance improvement of networks for reservoirs with a relatively large impoundment ratio is less significant, when adding extra information on storage, than the performance improvement of networks for reservoirs with a smaller impoundment ratio, which is in agreement with Hejazi et al. (2008).

The distribution of the impoundment ratios of the reservoirs in the GRAND database (Lehner et al., 2011) has a median impoundment ratio of 1.09, see Figure 16. Most of these reservoirs have a storage capacity larger than their yearly inflow. By extrapolating the effects observed in our limited set of reservoirs, it is likely that their potential fuzzy rules will be more dependent on reliable storage information than on the current or previous month's inflow.

For the case of adding a ToY parameter, see Figure 14b, it is easy to understand why this could help improve performance in theory. Management of reservoirs often anticipates the occurrence of dry and wet seasons by applying different modes of operation. The addition of this variable allows the fuzzy rules to make a clear distinction between seasons and the seasonality of flows. By evaluating the significance of improvements resulting from adding the ToY parameter as an input to a network, it becomes clear that there is not much value to this addition. In some cases, the addition of the ToY parameter results in significant improvements. These cases appear quite randomly, implying that the increase in rules and consequence parameters is responsible for the improvement, rather than the information added.

## 5.3 Applicability to GHMs

Implementation of ANFIS derived Fuzzy Rules into GHMs presents a challenge different from the ones posed by the more traditional simulation and optimization based algorithms. Mainly because of the need to acquire relatively extensive data on inflows, storage levels and release flows for each reservoir.

Nevertheless, the advent and expected development of Remote Sensing (RS) techniques to monitor water resources on a global scale allows for optimism and the proposed methodology provides opportunities to take full advantage of these developments. As shown by the Joint Research Centres Global Surface Water Dataset (Pekel et al., 2016) and Deltares Aqua Monitor (Donchyts et al., 2016a, b), water surfaces can be observed using freely available RS datasets. As both the spatial and temporal resolutions of newer RS products improves, the accuracy of these measurements can be expected to improve accordingly. By combining the spatial extends of water-bodies, water level measurements from altimeters and relations derived from a DEM (van Bemmelen et al., 2016) between these two indicators and a reservoirs volume, time-series of a reservoirs storage can be determined.

Subsequently, the inflows into a reservoir are needed to train a network. Simons et al. (2016) showed for the Red River basin in northern Vietnam how global RS datasets of precipitation and evapotranspiration can be combined to examine hydrological

processes like storage changes and stream-flows in small sub-catchments upstream of stream-flow measuring stations. They conclude that if storage changes are given, predictions of monthly stream-flows can be made. In analogy to their method, flows could be determined for sub-catchments of dams using the aforementioned estimations of reservoir storages. Since these estimates might not be as accurate as in-situ measurements or results from a hydrological model it is important to realize
that the network uses fuzzy classifications, like 'low' or 'very high', to describe the inflows. Alternatively, inflows could be determined by the model hosting the reservoir algorithm.

After determining a time-series of inflows and storages, the release can be determined by applying a mass balance to the reservoir. These three steps of determining storage changes, inflows and releases could then be applied to reservoirs that are located furthest upstream in a basin first, working downstream from there. This way, using the trained networks of the
upstream reservoirs, the inflow into the next reservoir could already include the anthropogenic effect on stream-flow of the upstream reservoir, mitigating the accumulation of errors between cascading reservoirs along a major river.

Alternatively, the system scale effects of cascading reservoirs can be dealt with by implementing a cluster of reservoirs as a single reservoir, represented by a single set of fuzzy rules. Fuzzy rules as described can represent these systems by defining the storage term as the sum of the individual reservoirs storages, the inflow as the inflow into the most upstream reservoir, and
the release as the release from the furthers downstream reservoir.

Once the data required for the training of a network has been acquired, the actual training is a straightforward and easily automated process, resulting in a calibrated network that can in a computationally cheap way quantify release decisions based on the inputs.

Although all the variables associated with the fuzzy rules have a physical basis, it is possible that a trained network releases
more water than is actually stored in its reservoir because the network does not keep track of a mass balance. Since simulated peak releases do not deviate much from the actual releases, see Figure 12, it is unlikely that a reservoirs storage becomes smaller than physically possible. Nevertheless, it would be necessary to keep track of a simple balance and bound the release to the water that is available in the reservoir, ensuring that never more water is released than has been stored in the reservoir.

Just like the more traditional generic operating rules, the proposed method will suffer from errors in the reservoirs in-
flows generated by the host model, errors due to the interdependence of cascading reservoirs and errors attributed to the non-stationarity of rule curves. As mentioned before, the errors in inflow are expected to be mitigated by the fuzzification, while the errors due to cascading could be restrained by incorporating the upstream anthropogenic effects of dams on inflows in the training set.

Regarding the the non-stationarity of rule curves, Jang (1993) already described a method to account for time-varying
characteristics of incoming data to the ANFIS network. By adding a 'forgetting factor' $\lambda$ to Equation 10, the influence of older training samples on the configuration of the network can decay:

$$S_{i+1} = \frac{1}{\lambda} \cdot \left( S_i - \frac{S_i \cdot a_{i+1} \cdot a_{i+1}^T \cdot S_i}{1 + a_{i+1}^T \cdot S_i \cdot a_{i+1}} \right) \tag{23}$$

Where $\lambda$ is chosen between 0 and 1. When $\lambda$ is 1, no decay occurs, while smaller values increase the decay of older samples.

However, the inter-annual variability of flows also needs to be reflected in the time series. Choosing a too short time frame in order too avoid issues with the non-stationarity of rule curves or applying a too strong forgetting factor can obstruct this. Possibly, the return period of hydrological droughts can be a good point of reference.

## 6   Conclusions and Recommendations

It has been shown that using fuzzy logic and ANFIS, operational rules of existing reservoirs can be derived without much prior knowledge about the reservoir. Their validity was tested by comparing actual and simulated releases with each other and by comparing the performance of the proposed method with a simulation based algorithm. The rules can be incorporated into GHMs or more regional models struggling with reservoir outflow forecasting. After a network for a specific reservoir has been trained, the inflow calculated by the hydrological model can be combined with the release and an initial storage in order to calculate the storage for the next time-step using a mass balance. Subsequently, the release can be predicted time-steps ahead using the inflow and storage.

Although adding the ToY to the mix of input parameters does not seem to result in significant improvements in release prediction, adding other input parameters might. Many macro-scale reservoir modelling algorithms use downstream water demands as input, which is a important factor in reservoir operating decisions. Adding this parameter would allow the fuzzy rules to describe operating decisions more accurately, especially for irrigation reservoirs.

More research on the optimal set-up of fuzzy rules per reservoir type is needed in order to get a better understanding of how the physical properties of a reservoir affect the results. It has been shown that set-ups with information on storage in previous months significantly improve results for reservoirs with small impoundment ratios. Similar tests should be done for different types of reservoirs, by splitting the reservoirs into groups based on their primary purpose, uncertainty of the available hydrological information or the local climate, this requires a larger set of reservoirs however. As shown by Hejazi et al. (2008), dam operators base their release decisions on different kinds of information for different types of reservoirs and a better understanding of these decisions could help improve the interpretation of the results.

Besides the extension of the neural network with new or extra parameters, the membership functions themselves also show room for improvement. In some cases, the shapes of the trained membership functions lead to the activation of multiple fuzzy rules for a single sample. This is undesirable, because it greatly undermines the basic principle of fuzzy logic. Input is translated into linguistic labels and processed by fuzzy rules which represent human behaviour and knowledge. When samples are processed by multiple rules, the logical interpretation of a network becomes much harder. Wismer and Chattergy (1978) propose a method called the constrained gradient descent in which some limitations with regards to the bell shaped function (see Equation 2) are formulated. Considering $\{a_i, b_i, c_i\}$ and $\{a_{i+1}, b_{i+1}, c_{i+1}\}$ and setting $c_i + a_i = c_{i+1} - a_{i+1}$ ensures that the sum of two consecutive membership functions never exceeds 1. Simultaneously, it is possible to set conditions such that membership functions cannot become zero over the entire input domain.

A drawback of applying the proposed method, compared to other macro-scale reservoir modelling algorithms, is the need to acquire in-situ time-series, which is often problematic as a result of multilateral mistrust (Alsdorf et al., 2007). The last

decade, the possibilities of observing reservoirs from space using altimeters and radar and optical imagery have grown fast and this trend is expected to continue as more satellites are scheduled for launch (van Bemmelen et al., 2016). Combining the method proposed here with remotely sensed time-series could further open possibilities for GHMs, by allowing the derivation of operational rules for most reservoirs around the world.

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

## Appendix A: Introduction to Fuzzy Logic

In Figure 1, the four steps of fuzzy logic are visualised. A storage of 520 Mm$^3$ and an inflow of 123 Mm$^3$/month is given as input. In this example, the storage can be either fuzzified through the membership functions as "low" or "high" and the inflow as "low", "medium" or "high". Note that the shape of the membership functions is triangular here, but many shapes are possible. For the given membership functions, the storage is only classified as "high", the inflow however is both "medium" and "high" (implying that, in practice, some operators would classify this inflow as "medium" and some as "high"). This means two fuzzy rules are relevant for the given input:

- `IF storage is high AND inflow is medium, THEN outflow is Z1`

- `IF storage is high AND inflow is high, THEN outflow is Z2`

The storage has been fuzzified, it is assigned the membership function "high" and its associated membership value is 0.8. Similarly, the membership values for a "medium" and "high" inflow can be determined. They are 0.6 and 0.4 respectively.

Now the firing strengths, giving an indication of the relative importance of each rule, need to be determined. This can be done in many ways. In this example, the membership values are multiplied with each other. For the first rule, the "high" storage has a membership value of 0.8, while the "medium" inflow has a membership value of 0.6. The firing strength of this rule is W1 = 0.48. In the same manner, it follows that the firing strength of the second rule is W2 = 0.32. Implying that, in general, more operators associate the current situation, the storage and inflow, with the first rule than with the second.

It is possible to describe the consequences of rules in many ways, in this example and study, they are linear combinations of the input variables as described by Takagi and Sugeno (1985):

$$Z = p \cdot storage + q \cdot inflow + r \tag{A1}$$

In which {p,q,r} are parameters to be determined when determining the fuzzy rules.

5      Finally, the consequences can be aggregated by using a weighted average to acquire the release:

$$release = \frac{W1 \cdot Z1 + W2 \cdot Z2}{W1 + W2} \tag{A2}$$