# Peer review of "Deduction of Reservoir Operating Rules for Application in Global Hydrological Models"

_Hydrology and Earth System Sciences, 2016_

## Short Comment (SC1) · 27 Jan 2017

The manuscript describes a data-driven approach for the zero- and one-month ahead prediction of the discharge from water reservoirs. The discharge is predicted by an Adaptive-Network-based Fuzzy Inference System (ANFIS), which requires information on reservoir storage, inflow (with different lags) and seasonality to issue a forecast. The ultimate goal is to use reservoir-specific data-driven models to simulate the storage and release dynamics of water reservoirs in global hydrological models. The results obtained on 11 reservoirs appear to be encouraging.

We read the paper with interest, and have some comments that might be useful to the authors.

[Figure]

1. The approach requires the availability of (sufficiently-long) time series of inflow, storage and release for all reservoirs considered in a regional or global hydrological model. This represents a challenge for the immediate use of ANFIS, since actual releases are generally not available. The use of remotely sensed data–as discussed at line 23-28 (page 23)–is an interesting idea, but has yet to be developed fully for application in this context. For this reason, we believe that optimisation-based algorithms are the most ready-to-use technology available for regional and global studies (Turner and Galelli, 2016).

2. The good performance presented in the manuscript are obtained by feeding ANFIS with measured storage and inflow data. As discussed at line 4-9 (page 23), ANFIS should be used within a simulation model, where the storage dynamics are simulated with a mass balance. Hence, there is a risk that the errors in the prediction of the release are integrated over time–as the simulation progresses–leading to an inaccurate simulation of release and storage dynamics.

3. The model building process (training + validation + test) might be biased by the presence of drier years in the validation set (see Figure 6). The problem could be solved by adopting another validation scheme, such as cross-validation, which allows to account for all available information during the calibration process.

4. The authors may find some useful information in the study by Hejazi et al. (2008), who studied how different hydrologic information affect release decisions in 79 reservoirs in California.

References

HEJAZI, Mohamad I.; CAI, Ximing; RUDDELL, Benjamin L. The role of hydrologic information in reservoir operation–learning from historical releases. Advances in water resources, 2008, 31.12: 1636-1650.

TURNER, Sean WD; GALELLI, Stefano. Water supply sensitivity to climate change:

An R package for implementing reservoir storage analysis in global and regional impact studies. Environmental Modelling & Software, 2016, 76: 13-19.

Stefano Galelli, Singapore University of Technology and Design, Singapore (stefano_galelli@sutd.edu.sg)

Sean W.D. Turner, Pacific Northwest National Laboratory, U.S.A. (sean.turner@pnnl.gov)
* * *

---

## Referee Comment (RC1) · Anonymous Referee #1 · 10 Feb 2017

Review of "Deduction of reservoir operating rules for application in global hydrological models" by H. Coerver, M. Rutten and N. van de Giesen for potential publication in HESS

The manuscript describes an approach to derive reservoir release rules at a monthly time scale based on inflow and storage observe information with the goal of improving upon generic operating rules and forward looking optimization schemes used in global hydrology – reservoir models presently. The fuzzy approach is applied over 11 reservoirs globally with sensitivity test on the fuzzy rules and predictors (shapes, inflow and storage across different time steps interdependencies).

Overall comments:

[Figure]

The approach and motivation of the paper are of high interest to the HESS community following up on previous large scale water modeling. There is a need for such an approach in order to improve upon the generic operating rules while within the constraints of an optimization without forward looking optimization. While the approach is very sound, it fails to evaluate the improvement upon generic operating rules .

The generic operating rules take into consideration the expected inflow, reservoir storage, seasonality of flow and release for water demand and environmental constraints. The generic operating rules are therefore calibrated for the specifics of each reservoir, using data available for all of them in a consistent manner. They also allow ensuring that constraints are met at a finer temporal resolution in addition to monthly release targets (spill, environmental flow). The rules also allow for inter-annual variability. The rules have been further improved with storage targets (Voisin et al. 2013), which improves the pattern of release which is goal and storage dependent oriented. The rules overall mimic the seasonality in regulation although do not necessarily follow the operational rules and there still could be large differences with respect to reality. The current approach explores the optimization of the rules based on observed inflow and storage, and tries to match observed releases, therefore could allow for a more realistic seasonality in the rule curves. However it is unclear how they improve upon the generic rules while global hydrologic modeling has been more focused on improving other physically based processes (groundwater) rather than generic operating rules ( Wada et al. 2016).

1/Applicability to GHMs : This paper presents an approach presently using the best case scenario (observed input data) and will likely lead to other evaluation within GHMs using GHMs flow and reevaluation. In order to meet a first objective of the paper which is to improve the representation of reservoir release in GHMs, I would recommend discussing the anticipated applicability in a GHM context:

o significant errors in inflow? o the cascade of errors in release between cascading reservoir along major rivers? o Lack of observed release for most reservoirs? o Isolate
the non stationarity in rule curves as more reservoirs and water uses were built during the inflow, release and storage observation periods?

Those points should be further discussed in the paper in order to support the approach and its application to GHMs despite presented here as a proof of concept.

2/ technical evaluation:

2.1. comparison with generic operating rules

Another objective is to demonstrate the improvement upon the generic operating rules - I would also suggest to make an explicit comparison with the operating rules. Those are simple enough the recreate using an excel table and could be derived using the 10 year training dataset and tested over the same two years. What is unclear is if a simple calibration of the generic rules parameters could outperform the fuzzy approach. Despite the shortcoming of the fuzzy approach (data centric, etc), is the improvement toward more realistic rule curves such that it should be implemented over as many reservoirs as possible, data permitting, while completed with generic rules?

2.2. longer evaluation approach in order to capture trends and insight based on flow seasonality and reservoir characteristics

The experimental approach consists in exploring the parameterization and variability in parameterization across multiple types of reservoirs. From a mathematical perspective it sounds very valid but the paper presently lacks in insight from a physical perspective and in particular what we already know or what we learn with respect to water management. For example, the manuscript mentions operational constraints such as end of the year carry over or storage targets as a major driver of rules. Yet the current approach does not seem to simulate the carry over storage target. GHM have been used to understand terrestrial water variations (Doell et al. 2012, Pokhrel et al. 2012, Wada et al. 2016). This is an important aspect that is not represented by the operating rules. The fuzzy approach does not take it into consideration either. I would suggest putting

some of the exploration discussion in supplemental material and add insight with respect to what has been done so far and the scientific and realism results contribution of the approach,

Specific comments:

Section 2: before into going into the technical methodology, describe how you anticipate the approach to complement or build upon previous approaches and how you will measure it, and address the scientific questions

Table 1: how were those reservoir selected out of the 6000 large reservoir globally?

Section 4: average performance is shown for 2 reservoirs. Please define average performance.

Döll, P., K. Fiedler, and J. Zhang (2009), Global-scale analysis of river flow alterations due to water withdrawals and reservoirs, Hydrol. Earth Syst. Sci., 13(12), 2413–2432, doi:10.5194/hess-13-2413-2009

Döll, P., H. Hoffmann-Dobrev, F. T. Portmann, S. Siebert, A. Eicker, M. Rodell, G. Strassberg, and B. Scanlon (2012), Impact of water withdrawals from groundwater and surface water on continental water storage variations, J. Geodyn., 59–60, 143–56, doi:10.1016/j.jog.2011.05.001

Pokhrel, Y. N., N. Hanasaki, P.J.-F. Yeh, T. Yamada, S. Kanae, and T. Oki (2012), Model Estimates of Sea Level Change due to Anthropogenic Impacts on Terrestrial Water Storage, Nature Geoscience, 5, 389-392, doi:10.1038/ngeo1476

Voisin, N., L. Liu, M. Hejazi, T. Tesfa, H. Li, M. Huang, Y. Liu, and L.R. Leung (2013), One-way coupling of an integrated assessment model and a water resources model: evaluation and implications of future changes over the US Midwest, Hydrol. Earth Syst. Sci., 17, 4555-4575, doi:10.5194/hess-17-4555-2013

Wada, Y., I. E. M. de Graaf, and L. P. H. van Beek (2016), High-resolution modeling of

human and climate impacts on global

---

## Referee Comment (RC2) · Anonymous Referee #2 · 13 Feb 2017

I read the manuscript with interest and I found it well written and clearly presented. I have only some issues that I think the authors should clarify:

1. On page 3, line 12-13, the authors say that fuzzy logic has not been used within the field of reservoir operation. Could you leave a word about this recent paper: Macian-Sorribes, H. and Pulido-Velazquez, M. (2016). Integrating Historical Operating Decisions and Expert Criteria into a DSS for the Management of a Multireservoir System. J. Water Resour. Plann. Manage., 10.1061/(ASCE)WR.1943-5452.0000712, 04016069

2. The description of the results (Section 4) is rigorous, but often too analytical. Physical meaning is a bit neglected, until we come to the discussion (Section 5). I would like to see the results presented in the lights of more physical links to the physical charac-

␣[Printer-friendly version]

[Discussion paper]

[Figure]

teristics of the watersheds and reservoirs. For instance, the relation between storage capacity and mean inflow (annual or even monthly flows, if one is focusing on the time step used) is only introduced at the end, while I think it is crucial to understand the performance of data-driven methods. Much of the results are explained in the lights of the length of the data time-series, but physical issues are also important and should be mentioned earlier. Also, for instance, how does the purpose of the reservoirs (irrigation, hydro-power, etc.) play a role (if any)?

3. In the 3-step "training/validation/test" procedure, I could not see the difference between "validation" and "test". From a first read, I had thought that "test was referring to using it in an "operational, real time" setup, where inflows were forecast/predicted by a model and used through the operation rules determined in the training/validation phase. I think however that I understood it wrong. Could you clarify this?

4. The issue of non-stationary data time series should be discussed given the context of the paper. What if the training period does not reflect the same conditions of the validation period? In an "ever changing world" (as supported by the Panta Rhei IAHS decade), there are strong chances that upstream catchment areas have changed in land use and occupation (not to mention climate changes) and that also other reservoirs have been built in upstream parts, influencing inflow. How does that affect the method applied and the results?

5. In Fig. 7 & Fig. 6, I would recommend to write the legend outside the plot, so the reader can have a full view of the simulations in the training period.

---

## Author Comment (AC1) · 14 Feb 2017

Dear Dr. Galelli,

First of all, thanks for the interest and the comments.

1.  We agree that currently less data intensive algorithms are the most ready-to-use technology available for regional and global studies. However, with the development of cloud-computing services, like for example the Google Earth Engine, we expect it will be become easier and more straight-forward to apply the method suggested on lines 23-28 (page 23) than it is now.

2.  The integration of errors in the prediction of the release over time is indeed a seri-

ous problem when implementing ANFIS into a simulation model. This problem could be overcome however, by correcting the storage once a new (remotely sensed) measurement of the actual storage is available. In case no correction is applied, indeed close attention should be paid to make sure the storage does not become unbounded (i.e. become very large or negative). Additionally, two simple rules could be added manually ("if storage = 0 -> release = 0" and "if storage = max -> release = max") in order to make sure the storage remains bounded between 0 and the maximum storage.

3. The use of cross-validation is a interesting suggestion, since indeed the presence of a dry year in the validation-set can undermine the results. As the length of the entire dataset increases however, the effects of a dry year will also decrease.

4. Hejazi et al. (2008) indeed looks like a very interesting and relevant study, thank you for pointing to it.

In the revised version of the article, we will make sure the answers to your comments are incorporated.

Regards, Bert Coerver

————————————————————

---

## Author Comment (AC2) · 25 Apr 2017

Dear referee,

Thank you for your comments and suggestions.

In case the current approach is implemented in a GHM, the ANN will be trained using inflow data derived from the model itself. Assuming that the variance in the errors in inflow values is not very large, the ANN will be trained with inflows containing the bias (opposed to actual observed inflows). By combining this inflow with remote sensing measurements of the storage in the respective reservoir, the release can then also be

determined.

The mentioned cascading effect can indeed cause problems. In case the reservoirs are close to each other and the operations are done in an integrated way, one could consider to lump all the reservoirs together and apply the ANN using the inflow in the most upstream reservoir and the combined storage of all the reservoirs.

Regarding the non-stationarity of the rule curves, it is possible to update the ANN online, giving a greater weight to more recent samples than older ones. This way the fuzzy rules will steadily adapt over time to new situations.

Table 1 shows the MSE and Nash-Sutcliffe (NS) coefficients for the selected dams of which the functions does not include irrigation modeled with Hanasaki et al. (2006), together with the indicators already presented in the manuscript. Comparing the indicators, it becomes clear that the proposed methodology performs better for five of the seven dams. While the remaining two perform similarly, with NS-coefficients of 0.70 compared to 0.54 for Charvak and 0.83 compared to 0.75. Therefor it would indeed be a good idea to implement the fuzzy approach over as many reservoirs as possible, data permitting, while completed with generic rules.

Considering your comment on the carry over storage, perhaps this is not clear enough from the manuscript, but the ANN can indeed simulate carry over storage. For the case in which the ToY parameter is applied, it is possible that the rules describing the release around the end of the year incorporate the behavior of the dam operator with regards to the carry over storage target. In case the storage is below the target during the last months, the release described by the rules for these specific months should reflect that.

Regards, Bert Coerver

**Table 1.** The test MSEs ($10^{-3}$) [-] and the NS coefficients [-] for all dams for different time-ranges and with different prediction horizons together with the indicators using the Hanasaki et al. (2006) method.

| Range | Lag | | Dam | | | | | | | | | | |
| --- | --- | --- | AJ | BL | CF | CD | CV | KR | NR | SN | TT | TQ | TM |
| 1 | 0 | MSE | 23.9 | 41.1 | 5.80 | 71.2 | 5.68 | 23.6 | 15.2 | 16.0 | 21.1 | 12.3 | 19.8 |
| | | NS | 0.69 | 0.46 | 0.80 | -0.49 | 0.92 | 0.45 | 0.78 | 0.40 | 0.33 | 0.50 | 0.95 |
| 2 | 0 | MSE | 5.10 | 15.8 | 1.85 | 4.13 | 32.3 | 6.27 | 3.31 | 11.6 | 9.60 | 6.18 | 0.981 |
| | | NS | 0.93 | 0.79 | 0.94 | 0.91 | 0.54 | 0.85 | 0.95 | 0.57 | 0.70 | 0.75 | 0.98 |
| 2 | 1 | MSE | 41.0 | 31.9 | 5.78 | 23.6 | 13.0 | 32.6 | 23.0 | 12.0 | 28.0 | 24.1 | 21.5 |
| | | NS | 0.46 | 0.58 | 0.80 | 0.51 | 0.81 | 0.23 | 0.66 | 0.55 | 0.12 | 0.01 | 0.5 |
| 2 | 2 | MSE | 46.6 | 41.5 | 21.5 | 48.3 | 30.7 | 115 | 40.2 | 21.9 | 39.1 | 50.8 | 34.6 |
| | | NS | 0.42 | 0.45 | 0.24 | -0.02 | 0.55 | -1.67 | 0.39 | 0.18 | -0.19 | -0.91 | 0.21 |
| Hanasaki et al. (2006) | | MSE | 21.9 | 48.9 | 6.34 | - | 13.2 | 15.2 | - | - | 28.6 | 7.57 | - |
| | | NS | 0.51 | 0.11 | 0.22 | - | 0.70 | 0.52 | - | - | 0.02 | 0.83 | - |

**1 References**

Hanasaki, N., Kanae, S., and Oki, T. (2006). A reservoir operation scheme for global river routing models . Journal of Hydrology, 327(12):22 41.

---

## Author Comment (AC3) · 25 Apr 2017

Dear referee,

Thank you very much for your comments and suggestions, I will take them into consideration in a final version of the paper.

Regarding your question about the "training/validation/test" procedure, it should be understood as follows. The data used is split into three "sets". During the training of the rules, the training and validations sets are used (I understand this is confusing and will try to clarify it in the final paper), while the data in the test set is not used during training

of the rules. The data in the training set is actually put trough the ANN, both backward and forward, in order to update the rules. Simultaneously, the performance of the rules is tested using the validation set. The results from these tests are used to determine when the training is finished. Then finally, the trained ANN is tested with the test set, to acquire a result independent from the training and validation set.

As also mentioned in the first referee's comment, the non-stationarity of the data time series is indeed an issue within the current methodology. In the ANFIS algorithm, there is a parameter present that can correct for changing conditions, by giving a greater weight to more recent training samples. This way, the rules slowly "forget" about samples that are too back in time. In the current study, this mechanism is not applied however and all samples have the same weight.

As long as the actual operating rules (for example as described by a rule curve) did not change, the fuzzy rules within the ANN should still be applicable. If the upstream hydrological conditions change, trough climate or landuse change for example, the ANN will more often activate the rules describing these more "extreme" circumstances then before, but that does not necessarily mean the consequences of these rules are wrong. Of course this does not hold if trough hydrological changes, circumstances arise that were simply not present in the training set. In that case it is theoretically impossible for the ANN to know what the response will be. So indeed, when applying this method in a GHM, the rules would have to be updated every so often.

Regards, Bert Coerver

---

## Author Response (AR1)

**Track of Changes Revision**

Deduction of Reservoir Operating Rules for Application in Global Hydrological Models
* * *
**#. Summary of comment**
**By Referee**

*Reply*

- **Adjustments in manuscript**
* * *
**1. Evaluate and compare upon improvement on generic operating rules.**
**By Referee #1**

*In order to show improvements upon generic operating rules I've added a simulation with the Hanasaki et al. (2006) (HNS) algorithm to the manuscript, on which many others have defined variations. Section 3.5 in the manuscript now shortly explains this algorithm.*

*Table 2 in the adjusted manuscript shows the MSE and Nash-Sutcliffe (NS) coefficients for the the simulated versus the actual release from non-irrigation reservoirs, together with the indicators that were already presented in the original manuscript.*

*Comparing the indicators, it becomes clear that the most accurate set-up for the proposed methodology (range of two, no lag), performs better for six of the seven reservoirs (as indicated by the bold numbers in the Table). Only for Charvak reservoir HNS exhibits better performance indicators.*

*Besides the quantitative comparison with HNS, I've added a comparison between several aspects of generic operating rules and the proposed method at the end of section 2.*

- **Added reference to Voisin et al. (2013) in section 2, page 3, line 25**
- **Added section 3.5 (Comparison with a macro-scale reservoir algorithm), to explain how a comparison with generic operating rules was made.**
- **Adjusted Table 2 to include results from Hanasaki et al (2006).**
- **Added line in section 4.1 to make comparison with HNS.**
- **Added line in section 4.2 to make comparison with HNS.**
- **Added line in section 6 on comparison with HNS.**
- **Added several paragraphs at the end of section 2 to evaluate improvement upon generic operating rules.**

2. Discuss anticipated applicability in GHM context, with regards to errors in inflows, cascading reservoirs, lack of observations of releases and the non-stationarity of rule curves.
By Referee #1 & #2

*In case the current approach is implemented in a GHM, the ANN can be trained using inflow data derived from the model itself. Assuming that the variance in the errors in inflow values is not very large, the ANN will be trained with inflows containing the bias (opposed to actual observed inflows). By combining this inflow with remote sensing measurements of the storage in the respective reservoir, the release can then also be determined.*

*The mentioned cascading effect can indeed cause problems. In case the reservoirs are close to each other and the operations are done in an integrated way, one could consider to lump all the reservoirs together and apply the ANN using the inflow in the most upstream reservoir and the combined storage of all the reservoirs. Alternatively, networks could be trained in order from upstream towards downstream, already implementing the anthropogenic effect of upstream reservoirs on the inflow to downstream reservoirs.*

*Regarding the non-stationarity of the rule curves, it is possible to update the ANN online, giving a greater weight to more recent samples than older ones. This way the fuzzy rules will steadily adapt over time to new situations.*

- **Added Section 5.3 (Applicability to GHMs), discussing how the proposed algorithm can be implemented in GHMs.**

3. Manuscript is too analytical / lacks insight from a physical perspective.
By Referee #1 & #2

*By splitting the considered reservoirs into two groups, with either smaller or larger than median impoundment ratios, the importance of the different input variables can be investigated in relation to the impoundment ratio. Figure 14 in the manuscript showed that for relatively small reservoirs, the information regarding storage levels was less important for the results than for the larger reservoirs. This was also found by Hejazi et al. (2008), who investigated which information sources play an important role in the decision process of dam operators.*

*Since Hejazi et al. (2008) investigated a larger number of reservoirs, they could do similar tests for different kinds of reservoir characteristics (then only impoundment ratios), like the local climate, the primary purpose and uncertainty in available hydrological information available to operators.*

- ***Added some emphasis on the inclusion of the effect of impoundment ratios on results in section 1.***
- ***Added a paragraph about Hejazi et al. (2008) in section 2.***
- ***Added recommendation to investigate relations between reservoir types and optimal fuzzy rule configuration.***

- *Added line on comparison with Hejazi et al. (2008) in section 5.2.*

**4. Discuss how approach is anticipated to improve upon previous approaches.**
**By Referee #1**

*Whereas previous approaches use databases like GRAND or ICOLD to determine several characteristics of dams, the proposed method opens the door for the inclusion of information derived from earth observation satellites. Allowing for a wider implementation of reservoirs in GHMs than the roughly 7,000 dams included in GRAND.*
*Furthermore, the proposed method incorporates the aspect of human behavior in dam operation decisions, as was also described by Hejazi et al (2008).*

- **Changed last paragraph of section 1, describing more clearly the steps taken in the study.**
- **Added two paragraphs in section 2, describing how method will complement existing methods.**

**5. Explain how the reservoirs were selected.**
**By Referee #1**

*The reservoirs were selected based on data available to the authors at the start of the study.*

- **Added a line in section 3.3 to explain why these 11 reservoirs were selected out of the + 40.000 reservoirs globally.**

**6. Better explain why Charvak and Andijan reservoirs are shown in more detail.**
**By Referee #2**

*The membership-functions for Charvak and Andijan nicely showed some effects the training can have on them (i.e. moving left/right, becoming more/less steep etc.), and their convergence curves also show two extremes among those of the 11 reservoirs. Also, the average of the MSEs for different set-ups for these reservoirs, shows that Charvak is a reservoir with relatively good results (#3 out of 11), while Andijan has relatively bad results (# 9 out of 11).*

- **Adjusted formulation on selection of the 2 reservoirs.**

7. Place a reference to Macian-Sorribes and Pulido-Velazquez, 2016 on line 12-13.
By Referee #2

*Indeed Fuzzy Logic has been used in decision support systems for dam operators, by "translating" the results from optimization models to a more understandable language. I've added the reference to the list with similar studies.*

- **Added reference to list of references with studies that used fuzzy logic to improve Decision Support Systems for dam operators.**

8. Explain more clearly the difference between training, validation and test-sets.
By Referee #2

*The data used is split into three "sets". During the training of the rules, the training and validations sets are used, while the data in the test set is not used at all during training of the rules.*
*The data in the training set is actually put through the ANN, both backward and forward, in order to update parameters of the network. Simultaneously, the performance of the rules is checked using the validation set. The results from these checks are used to determine when the training is finished and also which state of the network was the best.*
*I.e., the training is stopped when the MSE with regards to the validation set starts increasing again (indicating that the solution has converged), once this happens the state of the network is restored to the state for which the validation-MSE was lowest.*
*Then finally, the trained ANN is tested with the test set, to acquire a result independent from the training and validation set.*

- **Added a paragraph in section 3.4 to more clearly explain the difference between the three datasets used.**

9. Place the legends in Figure 6 and 7 outside the plotting area.
By Referee #2

- **Moved the legends outside the plotting area.**

[revised manuscript text omitted]

---

## Author Response (AR2)

Text by Anonymous Reviewer
➔ Response by Authors

Clarifications have been added and the comparison with respect to the generic rules is adding tremendously to the paper. I strongly support the fuzzy approach to improve the representation of reservoir regulation of river flows. Some conclusions with respect to the overall contribution of the approach (improvement upon GRAND database, application to GHMs) remain however still unsupported.

**1) Applications to GHMs**

**1.1) Significant errors in inflow**
**1.1.1)** The approach still needs some support for claiming that it can be applied to GHMs. GHMs can only be calibrated to a certain extent and in specific locations, not at 40,000 nor 6,000 dam locations.

➔ Although the calibration of 40,000 reservoirs is not a task to be underestimated, it is important to keep in mind that these calibrations can be done independently from each other and from the GHM itself. The time needed to train the network for any of the 11 reservoirs presented in this paper never took more than 5 minutes on an everyday PC (and sometimes shorter, depending on how many epochs were needed). After this training, the network is configured and using it is computationally very cheap.

➔ Rather than the calibration, we think the acquisition of the time series (described in the first four paragraphs of Section 5.3) needed for the calibration is the main challenge, but with cloud computing products, such as the Google Earth Engine, and the further development of Remote Sensing missions, we believe this is feasible. Recent developments along these lines within our group has shown tremendous progress in this direction over a short time frame.

➔ We have added the following paragraph after the mentioned four paragraphs in Section 5.3, commenting on the calibration process:

*Once the data required for the training of a network has been acquired, the actual training is a straightforward and easily automated process, resulting in a calibrated network that can, in a computationally cheap way, quantify release decisions based on the inputs.*

**1.1.2)** GHMs have significant errors in the seasonality and the mean annual balance of the river flows. Those errors get even larger when irrigation and other sector demand need to be represented. Reservoir characteristics are set, i.e. if there is a 10% overestimation of flow, some small reservoirs can only regulate as much as the set reservoir capacity allows for it. The current approach seems to assume that the errors in flow are not taken into consideration, or are very small. Page 25 second paragraph does not address the point of the errors in flow and how it could affect the decision in particular.

➔ This is indeed a point of concern for both generic operating rules and the method proposed here, which is addressed in the second-last paragraph of Section 5.3. Since the network normalizes and fuzzifies all the physical parameters, an error of 10% is unlikely to change the classification (e.g. low, medium, high) of the flow and is thus unlikely to have a large effect on the consequence of the applied rule.

➔ It is important to realise that with the proposed methodology, no reservoir capacity is set directly. Rather, it is derived from the storage time series used during training and if an

overestimation of the flow would exceed the amount the reservoir can regulate, the network is likely to just let this water pass through, i.e. the proposed method is not going to solve errors in flows, nor reinforce it.

➔ Also, as shown in Figure 14 in the manuscript, for reservoirs with small impoundment ratios, the performance of the release is more dependent on the storage information than on inflows. The bulk of the reservoirs in the GRAND database, have small impoundment ratios (see point 2.1 below). Furthermore, as shown in the bottom row of Figure 12a, completely omitting the inflows from the fuzzy rules still gives an average MSE of 0.050 [-] for all the reservoirs (including the reservoirs with large impoundment ratios).

**1.2) Mass balance**

P25 first paragraph, the system says that there is no mass balance check, which is a no-go for GHMs. In some areas with groundwater-surface water interactions that are still a challenge to simulate, reservoirs dry up in the simulation but not in reality. For node-based water resources management models, the inflow input into the system is typically bias corrected in order to apply the observed regulation rules. Based on the previous point, if there is a consistent under/overestimation, how can you ensure that you are not creating a source/sink of water in the system.

Either the approach needs more clarification, or/and a mass balance check is necessary to claim application to GHMs in particular, and for integration in any hydrology model in general.

➔ Indeed, as mentioned in the third-last paragraph in Section 5.3, the neural networks applied to the individual reservoirs in this paper did not perform a mass balance check, which is a no-go for GHMs.

➔ In case the method is implemented in a GHM, a mass balance needs to be applied to the respective reservoir to ensure that the reservoir is not creating new water. By keeping track of a water balance, it is possible to bound the release to the maximum possible release based on the storage in the reservoir (i.e. not more water than stored can be released).

➔ The last sentence in the mentioned paragraph has been adjusted to put more emphasis on the necessity of a mass balance check when implementing the method in a GHM:

*Nevertheless, it would be necessary to keep track of a mass balance and bound the release to the water that is available in the reservoir, ensuring that never more water is released than has been stored in the reservoir.*

**2) Physical/operational insight**

An improvement is the analysis by type of impoundment and some conclusions now relate to this classification. It really improves the paper and the analysis! There are remaining concerns:

**2.1**) The current conclusion based on below and above median level of impoundment presently does not support the conclusion because the median is based on the 11 selected dams. Based on GRAND or ICOLD database, if you were to derive a level of impoundment for the 6000 dams, where are the dams selected for this paper?

➔ This is a very valid point indeed; the graph below shows a histogram of the impoundment ratios for the reservoirs in the GRAND database. The 11 reservoirs considered in the paper vary from 1.04 to 7.46 (impoundment = yearly inflow / storage). So our reservoirs are relatively small compared to the pool of nearly 7000 reservoirs.

➔ For about half of the GRAND reservoirs the storage capacity is larger than the yearly inflow, the median impoundment is 1.093. Roughly the 80[th] percentile of the GRAND impoundment ratios is equal to the median value used to split the 11 considered reservoirs into two groups (i.e. 3.97).

➔ Thus, most of the reservoirs in the GRAND database are more like the reservoirs that depend strongly on storage information and less on information regarding recent inflows (i.e. as in Figure 14b).

➔ We have added the histogram showing the impoundment ratios for the GRAND reservoirs to the manuscript and adjusted the paragraphs in Section 5.2 dealing with discussion of the influence of the impoundment ratios on the best configurations of the networks as follows:

*The distribution of the impoundment ratios of the reservoirs in the GRAND database has a median impoundment ratio of 1.09, see Figure 16. Most of these reservoirs have a storage capacity larger than their yearly inflow. By extrapolating the effects observed in our limited set of reservoirs, it is likely that their potential fuzzy rules will be more dependent on reliable storage information than on the current or previous month's inflow.*

[Figure]

**2.2**) The authors have complemented the discussion based on Hejazi et al. (2008) which discusses the human decision making process based on reservoir characteristics.

There is still no discussion on the type of hydrological regime for which the fuzzy approach improves upon the generic rules. Even for specific level of impoundment, the seasonality in flow can affect drastically the performance of the generic rules, and the fuzzy rules.

For example, in the context of a reservoir containing 30% of the annual flow and the spring snowmelt is 60% of the annual flow, the variations in reservoir storage from the generic operating rules are certainly further from operations rules as they will drive to drying and uncontrolled spilling. When the reservoir can store only 10% of the mean annual flow, in the same situation, the generic operating rules have much lower impact on the overall regulation. This is the sum of all the small reservoirs that make an impact on the regulation.

➔ As mentioned in the third paragraph of Section 6, a better understanding of the behaviour of the proposed algorithm in relation to physical properties like its primary purpose, uncertainty of the available hydrological information, and the local climate (or seasonality in flow) is needed. However, a presently not available larger set of reservoirs is needed for that.

➔ Seasonality in flow should normally be clearly visible in the time series of inflows, used to train the network. By applying the ToY parameter to the network, the fuzzy rules (or operating rules of the dam) can vary per season.

➔ For the considered reservoirs, the addition of this parameter does not result in significantly more accurate releases, implying that, probably, seasonality of flow is not very important for the reservoirs considered here.

➔ The last paragraph of Section 5.2 has been adjusted to also mention the seasonality of flows:

*For the case of adding a ToY parameter, see Figure 13b, it is easy to understand why this could help improve performance in theory. Management of reservoirs often anticipates the occurrence of dry and wet seasons by applying different modes of operation. The addition of this variable allows the fuzzy rules to make a clear distinction between seasons and the seasonality of flows. By evaluating the significance of improvements resulting from adding the ToY parameter as an input to a network, it becomes clear that there is not much value to this addition. In some cases, the addition of the ToY parameter results in significant improvements. These cases appear quite randomly, implying that the increase in rules and consequence parameters is responsible for the improvement, rather than the information added.*

With this perspective, the generic operating rules offer an advantage in that they allow representing all the reservoirs, even the small ones (which can be created without the GRAND database), and with less data constraints than the fuzzy approach. In the discussion of the improvement upon the generic operating rules, the system effect of many small reservoirs should be mentioned even if it does not get evaluated. This discussion would provide more insights on the actual improvement of the approach - only the reservoir scale is evaluated but not the system scale. This is okay, but it needs to be mentioned.

➔ Indeed, only single, isolated reservoirs have been considered. The anthropogenic effects of upstream dams on river flow can definitely have an effect on what is happening at a downstream dam. However, as long as these anthropogenic effects are included in the training data, those effects are also included in the fuzzy rules of the respective reservoir.

➔ Alternatively, it would be possible to model a cascade of reservoirs with a single trained network, describing the whole cascade with one set of fuzzy rules. In that case the inflow provided to the network would be the inflow into the most upstream reservoir, the storage the total combined storage of the individual reservoirs and the release would be represented by the release from the most downstream dam.

➔ This alternative has been added to Section 5.3 by adding the following paragraph:

*Alternatively, the system scale effects of cascading reservoirs can be dealt with by implementing a cluster of reservoirs as a single reservoir, represented by a single set of fuzzy rules. Fuzzy rules as described can represent these systems by defining the storage term as the sum of the individual reservoirs storages, the inflow as the inflow into the most upstream reservoir, and the release as the release from the furthers downstream reservoir.*

**3) Improvement upon previous approaches/Grand database**

I am not convinced that the approach here allows to improve upon the GRAND or ICOLD databases. The fuzzy approach can be applied to the GRAND database, and future satellite-based emerging databases. And so does the generic operating rules approach as well. Both approaches need storage characteristics, and information on inflow, (and demand, reservoir purpose if possible, etc, as applicable).

I suggest removing this section (in order to avoid confusion and divergence from the main objective of the paper around the fuzzy roles and improvement wrt the generic rules).

Else you could clarify that both generic and fuzzy approach can be used with both GRAND, ICOLD and other databases. Generic rules need only long term mean monthly flow while the fuzzy approach need a longer training periods (time series instead of average) with possibly a longer time series when the inter-annual variability is large (and this point could benefit from being discussed).

- ➔ Simulation schemes, such as the schemes by Hanasaki et al. (2008), use *static input* from the GRAND and ICOLD databases (e.g. storage capacity, main purpose etc.) to derive operational rules.
- ➔ We do not use any data from the GRAND or ICOLD databases, nor do we attempt to improve upon them. Instead, we use *dynamic input* (storage and inflow time series) to derive operational rules.
- ➔ It is important to make sure that the training data reflect all the hydrological regimes that can occur at a certain site, especially when the inter-annual flow variability is large. Dry, wet and average years should be included in order for the network to be able to emulate the response (i.e. the release) for those situations.
- ➔ On the other hand, too long time series can also cause problems due to the non-stationarity of rule-curves as described in Section 5.1 and at the end of section 5.3. Therefore, the return period of droughts might be a good indicator for the ideal length of the time series and for the optimal value of the "forgetting factor" lambda mentioned in Equation 23.
- ➔ The last paragraph in Section 5.3 has been complemented with the following text:

  *However, the inter-annual variability of flows also needs to be reflected in the time series. Choosing a too short time frame in order too avoid issues with the non-stationarity of rule curves or applying a too strong forgetting factor can obstruct this. Possibly, the return period of hydrological droughts can be a good point of reference.*

The improvement with respect to previous approaches should likely focus on the new analysis quantifying the improvement with respect to the generic rules only, which is already provided.

**Minor 1)** The text refers to figure 11 then jumps to figure 15.

- ➔ Fixed

[revised manuscript text omitted]

---

## Author Response (AR3)

Comment:

P5 Last line: remove or adjust the statement that the water management is performed at a monthly time scale in most macro scale water management. i.e. change " monthly algorithm" to " monthly pre-release estimate" .

> -> removed the last sentence of paragraph 2 / P5 Last line, the preceding sentence has shifted to the end of p4 in the corrected manuscript.